# Synthetic eco-evolutionary dynamics in simple molecular environment

Luca Casiraghi[1†], Francesco Mambretti[2†], Anna Tovo[2], Elvezia Maria Paraboschi[3,4], Samir Suweis[2]*, Tommaso Bellini[1]*

[1]Dipartimento di Biotecnologie Mediche e Medicina Traslazionale, Università degli Studi di Milano, Via Fratelli Cervi, Segrate, Italy; [2]Dipartimento di Fisica e Astronomia, Università degli Studi di Padova, Padova, Italy; [3]Department of Biomedical Sciences, Humanitas University, Via Rita Levi Montalcini, Pieve Emanuele, Italy; [4]IRCCS, Humanitas Clinical and Research Center, Rozzano, Italy

**\*For correspondence:**
samir.suweis@unipd.it (SS);
tommaso.bellini@unimi.it (TB)

[†]These authors contributed equally to this work

**Competing interest:** The authors declare that no competing interests exist.

**Abstract** The understanding of eco-evolutionary dynamics, and in particular the mechanism of coexistence of species, is still fragmentary and in need of test bench model systems. To this aim we developed a variant of SELEX in vitro selection to study the evolution of a population of ~$10^{15}$ single-strand DNA oligonucleotide 'individuals'. We begin with a seed of random sequences which we select via affinity capture from ~$10^{12}$ DNA oligomers of fixed sequence ('resources') over which they compete. At each cycle ('generation'), the ecosystem is replenished via PCR amplification of survivors. Massive parallel sequencing indicates that across generations the variety of sequences ('species') drastically decreases, while some of them become populous and dominate the ecosystem. The simplicity of our approach, in which survival is granted by hybridization, enables a quantitative investigation of fitness through a statistical analysis of binding energies. We find that the strength of individual resource binding dominates the selection in the first generations, while inter- and intra-individual interactions become important in later stages, in parallel with the emergence of prototypical forms of mutualism and parasitism.

## eLife assessment

In this **important** study, the authors develop a promising experimental approach to a central question in ecology: What are the contributions of resource use and interactions in the shaping of an ecosystem? For this, they develop a synthetic ecosystem set-up, a variant of SELEX that allows very detailed control over ecological variables. The evidence is **convincing**, and the work should be of broad interest to the ecology community, leading to further quantitative studies.

## Introduction

A central effort in theoretical biology and ecology is to provide an effective description of the intimate, but often subtle, relationship between a given environment and the evolution of its ecosystem. In the case of simple environments without geographical isolation and physical barriers, as the one here considered, the emergence of species and phenotyping clustering (***Davis and Mayr, 1943***; ***Rundle and Nosil, 2005***; ***Keymer et al., 2012***; ***Gupta et al., 2021***) is generally considered an outcome of the competition for the limited available resources (***Dieckmann and Doebeli, 1999***; ***Pigolotti et al., 2007***; ***de Aguiar et al., 2009***; ***Anceschi et al., 2019***). The coexistence of stable species is ecologically understood in terms of 'niches', indicating the unique role and position that a particular species occupies within an ecosystem. According to the 'niche hypothesis' (***Chase and Leibold, 2009***; ***Peterson, 2011***; ***Anceschi et al., 2019***), biodiversity is limited by the number of 'niches' (or types of resources)

that are present since no two species can occupy the same niche indefinitely, as one would eventually outcompete the other (*Levin, 1970*; *Gupta et al., 2021*).

*Fitness*, which quantifies species reproductive success and thus also their relationship with the environment (*de Visser and Krug, 2014*), cannot generally be defined in a predictive way even in the most idealized systems (*Thurner et al., 2010*; *Wiser and Lenski, 2015*). Indeed, despite the efforts to identify simple case study systems, the evolution of populations formed by a variety of species remains difficult to model and to quantitatively characterize because of the inherent complexity of ecosystems and living beings, of the large number of potentially relevant variables of difficult access, and of the role of stochasticity (*Catalán et al., 2017*).

In this scenario, introducing new tools to explore and test eco-evolutionary models, concepts and interpretations appear as the best strategy toward new understanding (*Solé, 2016*). Along this line, various synthetic biological platforms have been proposed in the last years (*Ichihashi et al., 2013*; *Tizei et al., 2016*; *Parrilla-Gutierrez et al., 2017*; *Kauffman et al., 2018*; *Adamala and Szostak, 2013*; *Katla et al., 2023*) that exploit different principles and mechanisms, and focused on in vivo, in vitro, ex vivo, or in silico approaches. For example, in *Ichihashi et al., 2013*, a 'cell-like' model system is proposed, in which the evolution of one long genomic RNA (>2000 nt long) was investigated in detail under the action of a selective pressure ultimately provided by its biological meaning. Despite the tremendous simplification provided by this approach, the constructed artificial cell is still a complex system that includes ribosomes, lipids, translation factors, accessory proteins, tRNAs, and amino acids. While most of the synthetic biological platforms are based on the establishment of cell-like, compartmentalized systems involving complex biomolecular milieus and processes (*Tizei et al., 2016*), a few are devoted to investigate, with distinct strategies, the competition between two coexisting vescicle-based species competing for resources (vescicle-forming molecules) in a non-biochemical context (*Adamala and Szostak, 2013*; *Katla et al., 2023*). These articles demonstrate that typical evolution concepts, such as competition or resources and niche exclusion principle, can be applied to 'non-biological' systems.

We here propose a synthetic eco-evolutionary scheme, with no cell-like compartmentalization and no connection to the molecular biology of the cell: no coding sequences, no translation, no proteins involved. We consider an ecosystem formed by a large crowd (~$10^{15}$) of distinct molecular individuals interacting with each other and competing for survival in an environment with fixed resources, and we focus on the process by which selection and competition drive the emergence of dominating species. Specifically, we study the evolution of a pool of 50-base-long single-strand DNA oligomers with random sequences, a choice that ensures that in the initial solution each molecule is unique. The mechanisms of survival and mutual interactions are based on DNA hybridization. By exploiting

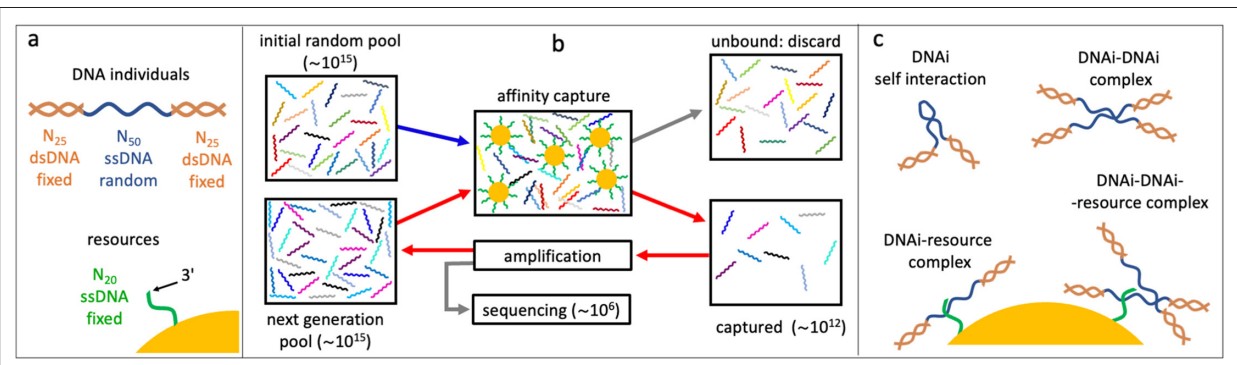

**Figure 1.** Affinity-based DNA synthetic evolution (ADSE). (**a**) Structure of the DNA oligomers participating in ADSE as individuals (DNAi) and target resources. (**b**) Steps in the ADSE. The process starts with a random-sequence DNAi population. The capture by magnetic bead-conjugated resources provides the selection: bead-bound DNAi are amplified to form the new generation, a small fraction of which is sequenced by massive parallel sequencing. The rest of the original solution is discarded. Red arrows mark the steps of each ADSE cycle. (**c**) Possible interaction motifs involving DNAi. The online version of this article includes the following figure supplements.

The online version of this article includes the following figure supplement(s) for figure 1:

**Figure supplement 1.** Detailed structure of DNA individuals.

**Figure supplement 2.** Detailed scheme of the affinity-based DNA synthetic evolution protocol.

the wealth of tools and knowledge about DNA code selective pairing and DNA synthesis, amplification and sequencing, we create a condition in which a limited number of accessible and computable variables control the destiny of an ecosystem formed by biological molecules that evolve in a non-biological way, i.e., with no reference to the biological meaning of their sequence.

The results of our work, demonstrating the intimate connection between fitness and ecological interactions, belong to the growing body of studies (*Fussmann et al., 2007*; *Vetsigian, 2017*; *Camacho Mateu et al., 2021*) showing that ecological and evolutionary processes are strictly related in the emergence and maintenance of species.

## Results
### Affinity-based DNA synthetic evolution

We introduce here a variant of SELEX for in vitro synthetic evolution of oligonucleotides to develop protein-binding aptamers (*Ellington and Szostak, 1990*; *Tuerk and Gold, 1990*). In standard SELEX protocol, the evolving oligonucleotides are selected at each cycle by their interactions of with the target protein. In the experiment reported here, we implemented a selective mechanism based on the affinity capture provided by magnetic beads carrying single-stranded DNA (ssDNA) filaments of fixed length $L = 20$ and sequence, that act as targets (or resources, *Figure 1a*). Selection is thus primarily based on the sequence of the DNA individuals and its level of complementarity to the targets. This marks a significant difference with SELEX, in which the aptamer-protein interaction depends instead on higher order factors such as the secondary structure of the oligonucleotides and the variety of binding sites on the folded protein. Being these hard to model, predict, and control, SELEX has never been considered, to the best of our knowledge, as a useful experimental test bench to understand evolution.

In our affinity-based DNA synthetic evolution (ADSE) protocol, evolution starts from an initial pool of DNA individuals (DNAi), chosen to be of fixed length $L = 50$ and random sequence. Each sequence indicates a 'species'. Since the potential molecular variety is $4^{50} \sim 10^{30}$ while our experiments use about $10^{15}$ initial molecules, each species in the initial pool has only one DNAi. The following evolution process is sketched in *Figure 1b*, *Figure 1—figure supplement 2*, and is given by three steps.

(i) *Selection*: the seed population is mixed with a given amount of dispersed capture beads. After a suitable incubation time, the beads are extracted from the solution and the bound oligomers released and saved. The rest of the original solution is discarded.

(ii) *Amplification*: the pool of 'survived' oligomers is PCR-amplified about 1000 times to recover the initial molarity.

(iii) *Sequencing*: a small portion of the amplified sample ($1–3 \times 10^6$ molecules) is analyzed by massive parallel sequencing. These molecules are thus removed from the evolving pool. These steps constitute a cycle - one generation of evolution - that we repeated up to 24 times in two independent evolution histories, which we refer to as 'Oligo1' and 'Oligo2'. In the following, we present in detail the results of the Oligo1 evolution, while Oligo2 results are described in the figure supplements.

In actuality, to enable amplification and sequencing, DNAi are built by flanking the 50mer with two 25-base-long fixed sequences that enable primer binding for a total length of 100 bases. Such two terminal segments of the DNAi are made inactive during selection by hybridization with oligomers of perfect complementarity, as sketched in *Figure 1a*, *Figure 1—figure supplement 1*.

A key feature of ADSE is that DNAi can interact not only with the resources, but also with itself and with each other, and also potentially form complexes binding to the affinity beads as summarized in *Figure 1c*. Indeed, the choice of a length of 50 for the DNAi interaction was thought to enable - in principle - simultaneous resource and mutual binding.

PCR amplification can be of high fidelity or error-prone, the latter choice enabling genetic drift and, potentially, speciation. Since our primary goal was to first investigate fitness in a pool of competing species within a given niche, defined in our case by the 20-base-long resources, we opted for a high-fidelity amplification, and left the investigation of high mutations regimes to a follow-up experiment. Because of the many PCR cycles required in the ADSE scheme, the amplification inevitably leads to the formation of artifacts in the form of longer sequences (*Tolle et al., 2014*), a phenomenon that intrinsically sets a limit to the number of generations that can be explored (see Appendix 1).

As detailed below, the ADSE protocol enables observing a non-trivial evolution of the DNAi ecosystem across generations, the emergence of dominating species and non-monotonic population evolution. It also enables to appreciate the role of inter-specific interactions and their contribution to fitness.

## Evolution of the DNAi ecosystem

The main output of the synthetic evolution that we are proposing is the dataset $\{DNAi\}_j$ obtained by sequencing DNAi at the various cycles ($1 \leq j \leq 24$ being the index of the cycle). We find that the initial random ecosystem markedly changes with ADSE generations as shown in *Figure 2a*, in which we have plotted the evolution of: (i) the fraction $F_D$ of the total population formed by distinct nucleotide sequences (red dots). $F_D$ drops from 1 to nearly 0, indicating that initially DNAi are all different from each other, while after 24 generations the number of distinct sequences is much smaller than the number of DNAi. This indicates that most of the initial sequences become progressively 'extinct'. (ii) The fraction $F_{10}$ of the total population formed by the 10 most abundant DNA sequences (black dots). $F_{10}$ is initially close to 0 - being each sequence represented by a single DNAi, and grows to about 25% at cycle 12 and 55% at cycle 24 - indicating that the offspring of 10 initial 'mother' sequences becomes the majority of the system. This loss of diversity can also be quantified by the zipped file size of the list of sequences (*Benedetto et al., 2002*) or by computing its Shannon entropy (*Hill, 1973*), both markedly decreasing during evolution (see Appendix 1).

Since in ADSE survival depends on bead capture, we expect DNAi-resource binding strength to be an essential ingredient of fitness. To this aim we compute the mean DNAi-resources binding free energy $\Delta G_{DR}$ across generations, plotted in *Figure 2b* (blue squares, right-hand side $y$-axis). Specifically, $\Delta G_{DR}$ has been computed based on DNAi and resource sequences by using the standard 'nearest-neighbor approximation' for the thermodynamics of DNA hybridization (*SantaLucia, 1998*; *Ghosh et al., 2019*; *Plata et al., 2021*), as implemented in the NUPACK tool in Python (*Zadeh et al., 2011*). Being this computation of some complexity, we could perform it only on batches of 1000 DNAi randomly chosen within $\{DNAi\}_j$, the error bars expressing the uncertainty introduced by such downsampling (see Materials and methods section). For reference, a DNA 20mer perfectly complementary to the resource would bind to it with an energy of approximately –24 kcal/mol.

To perform a faster - and thus more complete - analysis of the resource-binding energy in the evolving population, we have introduced $\omega$, a simpler quantifier than $\Delta G_{DR}$. $\omega$ is the length of the longest consecutive number of bases within each DNAi that is complementary to the resource sequence (considering all possible relative positions of the two), in which we allow for single pairing errors and 1-base bulges (*Mambretti et al., 2022*). $\omega$ ignores factors that are relevant for the free energy, such as the specificity of the sequence and the fraction of CG pairs, and thus is inadequate to evaluate the binding strength of specific DNAi. However, its average value $\langle \omega \rangle$ computed for the entire population in each generation (*Figure 2b*, red dots, left-hand side $y$-axis) grows almost identically to $\langle \Delta G_{DR} \rangle$ (blue dots, right-hand side $y$-axis). The $\langle \omega \rangle$ axis has been scaled so that its value (computed on a pool of random-sequence DNAi - dashed red line) matches, in the plot, $\langle \Delta G_{DR} \rangle$ (computed on the same pool - dashed blue line). A more detailed comparison between $\omega$ and $\Delta G_{DR}$ is given in *Figure 2c*, showing box plots with $\Delta G_{DR}$ on the $y$-axis and $\omega$ expressed through color code. Both are computed from a random selection of 1000 distinct DNAi from the initial (left-hand side box), the final populations (central box), and the top 10 most frequent sequences in the final generation (right-hand side box). The result further strengthens the validity, in our statistical context, of $\omega$ as a quantifier of the strength of interaction with the resources (*Mambretti et al., 2022*).

*Figure 2b* shows that, during ADSE, $\langle \omega \rangle$ tends to saturate at a value $\omega_{sat} \approx 10$, indicating that no relevant selective advantage is gained when $\omega > 10$, in agreement with the notion that the residence time of hybridized oligomers becomes larger than typical experimental times when $\omega > 12$ (*Di Leo et al., 2022*).

*Figure 2d–f* describes the evolution of the ecosystem $\{DNAi\}_j$ by showing $P(\omega)$, the fraction of DNAi having overlap $\omega$ with the resource evaluated in the initial pool (panel d) and at generation 12 (e) and 24 (f). $P(\omega)$ clearly evolves, its small $\omega$ components being progressively lost, while individuals with large $\omega$ grow in number, as they are more successful in being selected and amplified. It might be worth pointing out that the appearance of non-zero $P(\omega > 13)$ at generation 12, while $P(\omega > 13) = 0$ in the initial distribution, is an effect of the under-sampling involved in the sequencing procedure.

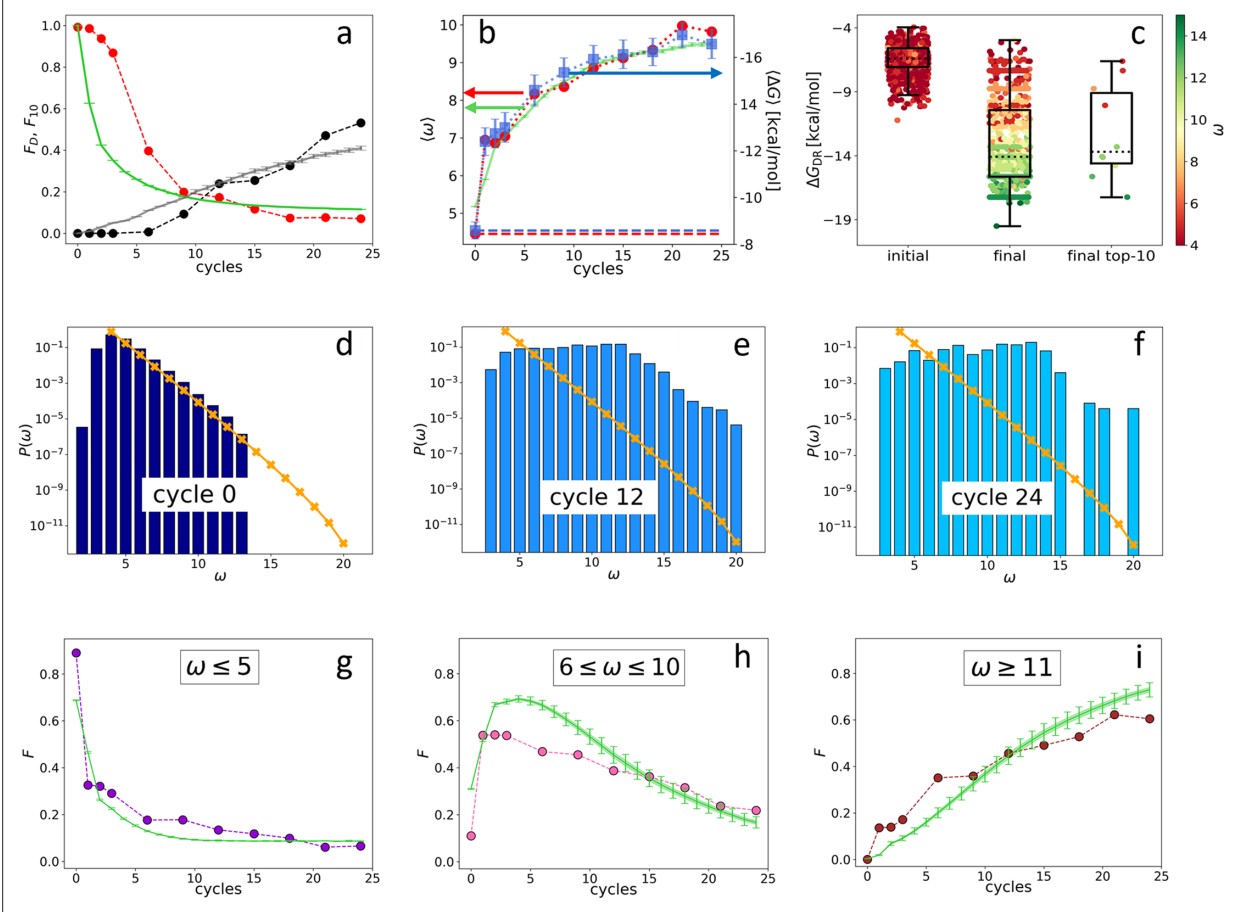

**Figure 2.** Evolution of the DNA individuals (DNAi) population (Oligo1 data). Time is expressed in affinity-based DNA synthetic evolution (ADSE) cycles. (a) Fraction $F_D$ of the total population formed by different sequences obtained from the experiment (red dots) and computed with the individual-based eco-evolutionary (IBEE) model (green line), fraction $F_{10}$ of the total population formed by the 10 most abundant sequences (experimental: black dots, IBEE model: gray line). (b) $\langle\omega\rangle$ computed on the whole population in each generation (red dots, left-hand side y-axis); $\langle\Delta G_{DR}\rangle$ computed on a sample of 1000 randomly chosen DNAi from the population in each generation (blue dots, right-hand side y-axis); data fitting with the IBEE model (green line, left-hand side y-axis). The left and right y-axes were scaled so that $\langle\Delta G_{DR}\rangle$ and $\langle\omega\rangle$ computed on a pool of random-sequence DNAi would coincide (dashed blue and red lines, respectively). (c) Boxplots and scatter plots of $\langle\Delta G_{DR}\rangle$ in ensembles of 1000 random sequences (left), 1000 randomly chosen DNAi extracted from the experimental population at cycle 24 (middle) and from the top 10 most populous DNAi at the same cycle (right). The color code is assigned to each point based on its $\omega$ value (color bar). (d–f) Probability distributions $P(\omega)$ for cycles 0 (d), 12 (e), and 24 (f). In the latter histogram, empty bins result from subsampling. Orange points and lines are the distributions evaluated with the null model. (g–i) Evolution of the abundance (expressed as fraction of the total population $F$) of sequences whose $\omega$ is small ($3 \le \omega \le 5$ - panel g), medium ($6 \le \omega \le 10$ - panel h) and large ($11 \le \omega$ - panel i) as obtained from the experiments (dots) and with the IBEE model (green lines). The model results are averages over 20 simulations. The online version of this article includes the following figure supplements.

The online version of this article includes the following figure supplement(s) for figure 2:

**Figure supplement 1.** Time evolution of the fraction of PCR by-products and of their $\langle\omega\rangle$.

**Figure supplement 2.** Probability distributions $P(\omega)$ of PCR by-products at cycles 0 and 24.

**Figure supplement 3.** Compared abundance of DNA individual (DNAi) populations between two sequencing replicates of the same library (cycle 9 of Oligo1).

**Figure supplement 4.** Whisker plots showing the fraction of 10 most abundant individuals $F_{10}$ for cycle 9 (red), cycle 9 replica (blue), and cycle 10 (green).

**Figure supplement 5.** $\langle\omega\rangle$ as a function of the experimental cycles for Oligo2.

**Figure supplement 6.** Initial (left, cycle 0) and final (right, cycle 18) $p(\omega)$ distributions for Oligo2.

**Figure supplement 7.** Evolution of the zip ratio CR for the file with the list of sequences (red dots), normalized by its value at time 0; the same for the individual-based eco-evolutionary (IBEE) model (green line); and of the Shannon entropy associated to the RSA distribution (experimental: black dots, IBEE: gray line).

*Figure 2 continued on next page*

*Figure 2 continued*

**Figure supplement 8.** Experimental vs IBEE time evolution of $\langle \omega \rangle$, for two different IBEE hyperparameters choices.

**Figure supplement 9.** Experimental vs IBEE time evolution of $\langle \omega \rangle$, for different sizes and compositions of the starting population of IBEE model.

The different destiny of DNAi with distinct $\omega$ values is shown in *Figure 2g–i*, where we show the temporal (i.e. across generation) evolution of the fraction of the total population whose $\omega$ is in the following ranges: $3 \leq \omega \leq 5$ (panel g), $6 \leq \omega \leq 10$ (panel h), and $11 \leq \omega$ (panel i). As expected, DNAi with weak affinity to the resources decrease, while those with large affinity increase. Interestingly, DNAi with intermediate affinity exhibit a non-monotonic behavior, indicating that the conditions for survival change during the evolution, reflecting the evolution of the ecosystem.

## Null model and eco-evolutionary algorithm

In order to better understand the experimental outcomes, we first build a null random model without evolution and then an individual-based eco-evolutionary (IBEE) model enabling predictions for $P(\omega)$.

The null model describes the interaction between individuals in the initial pool $\{DNAi\}_{j=0}$ of random sequences and the resources within a purely combinatorial framework (see Materials and methods and Appendix 3 for mathematical details). In this model, we attach a random string of length 50 to the resource string in a random position, and we compute the maximum consecutive overlap (MCO), accepting the binding only if it is at least formed by three complementary basis.

The resulting analytical $P_0(\omega)$, plotted in *Figure 2d* (orange crosses), closely matches the data obtained from sequencing, confirming the random-sequence nature of $\{DNAi\}_{j=0}$. The analytical $P_0(\omega)$ extends to a $\omega$ range where no data are available due to the limited size of the sequenced pool. The comparison between $P_0(\omega)$ and $P(\omega)$ in panels e and f shows that the exponential decay of $P_0(\omega)$ at large $\omega$ is maintained even in generation 12 and partly in generation 24, further supporting the notion that the selective advantage of $\omega$ saturates at large $\omega$.

In the IBEE algorithm we consider $N_p$ individuals. Each has a fitness $f_i(\omega)$ ($i = 1, 2, .., N_p$) that depends on its affinity $\omega$ with the resource. At time $t = 0$ we assign $\omega$ to each individual based on $P_0(\omega)$ from the null model. Then, for each evolutive cycle, we model competition and selection so that out of $N_p$ individuals, only $N_r$ survives. This is attained as a combination of two processes: a fraction $x$ of the $N_r$ sequences results from sampling individuals from the $N_p$ population, each with a survival probability $f(\omega) \in (0, 1)$ (selection); the remaining fraction $1 - x$ is extracted completely from the $N_p$ individuals. The latter group is meant to mimic 'neutral drift' (as it would be expressed in evolutionary language) provided by non-specific binding to the beads. The $N_r$ survivers are amplified by identical copying back to $N_p$, the starting population of the next evolutionary cycle (the introduction of very rare mutations does not affect the results). We perform 24 evolutionary cycles.

As expected, the outcome of the eco-evolutionary dynamics strongly depends on the shape of the fitness function. We model it as:

$$f(\omega) = \left( \frac{\omega^*}{\omega_{max}} \right)^{\gamma}$$

where $\omega_{max}$ is the (cycle-dependent) largest value of $\omega$ within the actual population and, $\omega^* = \omega$ if $\omega < \omega_{sat}$, while $\omega^* = \omega_{sat}$ otherwise. $\omega_{sat}$ and $\gamma$ are parameters to be tuned in the comparison with the observed $\langle \omega \rangle$. $\omega_{sat}$ expresses the loss of fitness gain for $\omega > \omega_{sat}$ yielding the saturation of $\langle \omega \rangle \cdot \gamma$ represents the strength with which $f(\omega)$ depends on $\omega$, and thus the rate at which low $\omega$ individuals are discarded. $\gamma$ may of course depend on time on account of the evolving ecosystem.

After grid search we find $x = 0.9$, indicating that the random drift contributes to about 10% of survival at each cycle, and $\omega_{th} = 10$, in agreement with the observations prompted by the saturating behavior of $\langle \omega \rangle$. We also find that the data cannot be approximated with a single value of $\gamma$, as visible in *Figure 2—figure supplement 8*. Data can instead be very well matched assuming $\gamma = 3$ for the first five cycles and $\gamma = 1$ for the remaining cycles, as shown in *Figure 2b*, green line. With the same choice of parameters, the IBEE fitness-based model captures the decrease in the diversity of the DNAi ecosystem (*Figure 2a*, green and gray lines) and the temporal (i.e. across generations) evolution of the relative population abundance of DNAi in the three $\omega$ intervals in *Figure 2*, panels g, h, and i.

The effect of the system size and initial conditions on the IBEE results, discussed in Appendix 2, do not qualitatively change the models results. Error bars on simulations have been obtained by averaging 20 independent runs, starting from the same initial conditions. As can be observed, the variability among simulations is negligible.

Also, the IBEE model with a fixed $\gamma$ would respect the increasing, decreasing, and non-monotonic trends of the data in *Figure 2*, but not quantitatively. The change in $\gamma$, and thus in fitness, is indeed necessary to reproduce the observed $\langle \omega \rangle$ with the IBEE model. This is a key result of our investigation because it indicates that, even in simple conditions of ADSE environment (fixed resource and low mutation rate), survival is controlled by more than affinity to the resources, as discussed in the analysis below.

## Self and mutual DNAi interactions are evolutionary drivers

While $\omega$ is certainly a key driver of the observed evolution, it is clearly not the only one. The fact that the top 10 most represented sequences become, in the last cycle, about 55% of the total population (*Figure 2a*) implies that, even among sequences with large $\omega$, only a tiny minority come to dominate, while the largest part of them eventually disappear. Moreover, the 10 most represented sequences do not stand out for their particularly large $\Delta G_{DR}$ or $\omega$, as noticeable in *Figure 2c*. The same data are plotted in *Figure 3a* as a $P(\Delta G_{DR})$ distribution to enable comparing the free energy distribution in the initial population (purple shading), in the final population (blue columns), and in the top 10 (black columns). These elements support the notion that survival and dominance must also be due to factors in addition to $\omega$, which we thus explored.

*Figure 3b* shows the value, across the 24 generations, of two other contributions to the total free energy, normalized to their value computed in random sequences, so to enable comparing their relative variations: $\langle \Delta G_{\text{self}} \rangle$ (green symbols), the average unimolecular free energy, expressing the average strength of the internal folding of each DNAi; $\langle \Delta G_{DD} \rangle$ (red dots), the total bimolecular DNAi-DNAi free energy, comprising both self and mutual interactions. For comparison, we also plot the normalized value of $\langle \Delta G_{DR} \rangle$ (blue dots). As for the case of $\langle \Delta G_{DR} \rangle$, $\langle \Delta G_{\text{self}} \rangle$ and $\langle \Delta G_{DD} \rangle$ are computed by randomly selecting 1000 individuals or pairs, respectively, from the population at cycle $j$ and using the NUPACK tool to compute the values.

*Figure 3b* reveals that $\langle \Delta G_{\text{self}} \rangle$ grows in time even more than $\langle \Delta G_{DR} \rangle$, but with a different progression: $\langle \Delta G_{DR} \rangle$ grows faster in the first cycles to later saturate, while the growth of $\langle \Delta G_{\text{self}} \rangle$ is more uniform.

Since $\Delta G_{DR}$ is computed as the free energy of the whole DNAi-resource structure, it includes contributions of self-energy associated to hairpins in the DNAi. Thus, the growth of $\langle \Delta G_{DR} \rangle$ could actually depend on the growth in $\langle \Delta G_{\text{self}} \rangle$ (but not the contrary). To investigate this possibility, *Figure 3c* displays the scatter plot between $\Delta G_{DR}$ and $\Delta G_{\text{self}}$, for a randomly chosen subset of DNAi at $j = 24$. In evidence are the 10 points corresponding to the 10 most populous sequences. The plot shows weak or no correlation, and a relevant shift with respect to $\Delta G_{\text{self}} = \Delta G_{DR}$ (black line), demonstrating negligible dependence of $\langle \Delta G_{DR} \rangle$ on $\langle \Delta G_{\text{self}} \rangle$ and indicating that these two quantities reflect two independent driving forces in the ADSE selection mechanism. The existence of two different growth regimes suggests that in the first stages of ADSE the selection is mainly dominated by affinity with the resources, while in later generations the requirement of stronger unimolecular folding becomes more important.

A similar scatter plot analysis for $\langle \Delta G_{DD} \rangle$ yields a different outcome. *Figure 3d* compares the $\Delta G_{DD}$ computed for two selected DNAi (individual '$k$' and '$l$') and the sum of $\Delta G_{\text{self}}$ of the same two DNAi. The apparent correlation indicates that, *on average*, a large part of $\langle \Delta G_{DD} \rangle$ simply embodies the growth of self-energy, although the difference $\Delta\Delta G \equiv \Delta G_{DD,kl} - \Delta G_{\text{self},k} - \Delta G_{\text{self},l}$ (orange arrow) is non-negligible. *Figure 3b* shows the behavior of $\langle \Delta\Delta G \rangle$ across cycles (orange squares). Despite the resulting mild growth of $\Delta\Delta G$ might appear not relevant, it actually indicated that ADSE, independently of $\omega$, selects strings that have higher reciprocal affinity than a random DNAi set (see Appendix 2). Indeed, the selection process could instead have promoted a decrease of the same quantity. It should be noted that mutual interactions might also involve the unfolding of hairpins self-structures, in which case their strength is much larger than $\Delta\Delta G$.

Self-interactions and mutual interactions can compete with the binding of DNAi to the resources either when they involve the same nucleobases or through steric hindrance. Therefore, we could

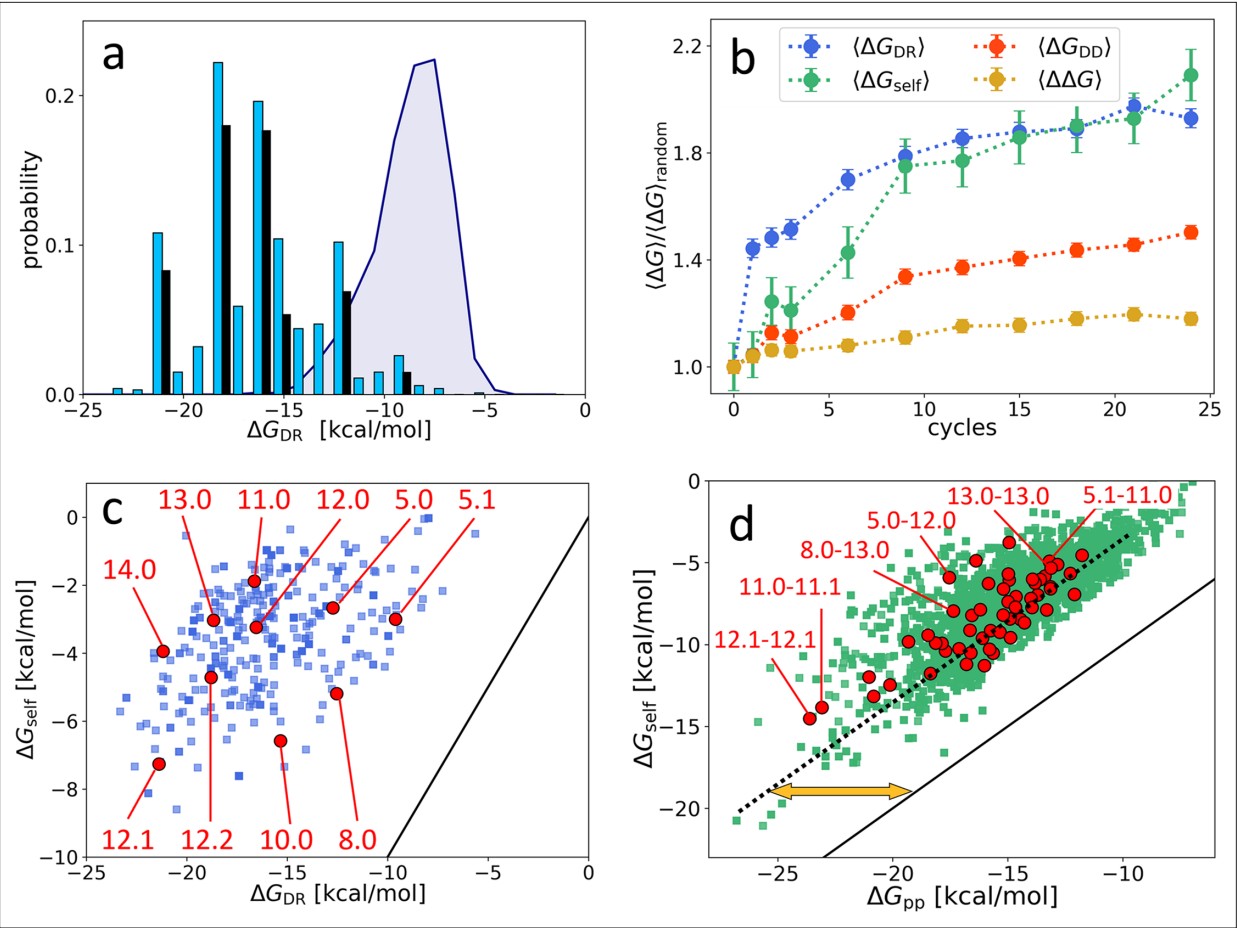

**Figure 3.** Distribution and evolution of free energy quantifiers (Oligo1 data). (**a**) Probability distribution for the DNA individuals (DNAi)-resource binding free energy computed by NUPACK, $p(\langle \Delta G_{DR} \rangle)$, for the initial population (gray shading), for the final population (random choice of 1000 DNAi - cyan columns, top 10 most populous sequences - black columns). (**b**) Time evolution, expressed in cycles, for various mean free energies $\langle \Delta G \rangle$, normalized to their value computed on pools of random sequences. All $\langle \Delta G \rangle$ are computed by NUPACK on sets of 1000 individuals: $\langle \Delta G_{DR} \rangle$ (blue dots); unimolecular self-interaction $\langle \Delta G_{self} \rangle$ (green dots); bimolecular mutual DNAi interaction $\langle \Delta G_{DD} \rangle$ (red dots); mutual, self-subtracted interaction $\langle \Delta\Delta G \rangle$ (yellow dots). (**c**) Scatter plot of $\Delta G_{self}$ vs. $\Delta G_{DR}$ computed for 1000 DNAi in the final population (blue squares). Red dots mark the point relative to the 10 most populous sequences, as identified by the labels. Note that the x-axis scale of panels a and c is the same, enabling identifying sequences. (**d**) Scatter plot computed on $10^4$ DNAi pairs from the final population comparing $\Delta G_{self,j} + \Delta G_{self,l}$ and $\Delta G_{DD,kl}$ (green squares). Red dots mark the pair formed by the 10 most populous sequences, some of which identified by labels. With respect to the condition $\Delta G_{DD,kl} = \Delta G_{self,j} + \Delta G_{self,l}$ (black line), data are on average displaced by $\Delta\Delta G \sim 7.5$ kcal/mol (yellow arrow). The online version of this article includes the following figure supplements.

The online version of this article includes the following figure supplement(s) for figure 3:

**Figure supplement 1.** Three examples of different relative positions for the attachment.

**Figure supplement 2.** Top: scheme of the target strand (red) and of a longer consumer strand (blue).

**Figure supplement 3.** Null model without threshold, analytical and simulated, with an even and uneven nucleotides distribution.

**Figure supplement 4.** Null model with/without threshold, simulated and analytical.

expect that the increase of $\Delta G_{DR}$ would lead to a decrease of both $\Delta G_{self}$ and $\Delta G_{DD}$. Their unforeseen growth in the ADSE process is hence an indication of the selective advantage they convey. We interpret this behavior as an indirect sign that mutual DNAi interactions, more than just an impediment, are major deadly threats for their survival. To avoid them, DNAi need screening. Indeed, the strong growth of self-interactions, and the mild increment in mutual interactions, could represent the emergence of 'defensive' strategies, as discussed below.

In our search for evolution quantifiers other than resource-binding strength, we also found that selection favors binding to resource sequences close to their resource 3' terminal, away from the

bead surface (see *Figure 4—figure supplement 2*). While this is expected, being that terminal less constrained and in a less crowded environment, it also provides useful clues to further analysis.

## The evolution of DNAi species

We examined the evolution of DNAi species, i.e., the change in numerosity of the groups of DNAi having equal sequence, which we report in *Figure 4a* relative to a limited set of (mainly) successful species (for a larger selection, see SI). Data in *Figure 4a* are expressed as the fraction $F$ of the total population that belong to a given species. It can be noticed that species that are later becoming dominant 'appear' in our analysis only at generation 5 in a single or a few copies. This is reasonable since we sequence a sample of ~$10^6$ DNAi over a much larger population of ~$10^{12}$ survived individuals.

Species are named after their $\omega$ value, combined with an index (e.g. '12.0', '12.1', '12.2'…) expressing their ranking in populousness. In *Figure 3*, panels c and d, we marked and labeled in red the free energy values associated to the 10 species dominating in the last generation and labeled the corresponding species or pairs of species. It can be noticed that the dominant species are not - as one could naively expect - at the extremes of the energies distributions and assigned, confirming that the statistical trends do not enable, by themselves, a full understanding of ADSE.

The populousness of ADSE species does not have standard patterns of evolution. This is true even when limiting the observation to the successful species in *Figure 4a* whose population growths are varied and do not reflect in any simple way the energy quantifiers. Conversely, species with similar energy quantifiers may have opposite fate, either becoming dominant or rapidly extinguishing or else displaying a non-monotonic population evolution. It is worth pointing out that all species start equal, each represented in the original pool by a single molecule, which on this scale would appear as a $F \approx 10^{-12}$ (black dot in *Figure 4a*, right panel).

We thus decided to seek further understanding on the survival of the fittest in ADSE by inspecting the 'natural history' of a few species (see list of sequences in the additional files).

13.0 and 11.0 (orange and gray dots in *Figure 4a*) are, respectively, the most and second most populous species in the last generation. It might be relevant to mention that the replica of ADSE described in the SI leads to different dominant sequences, as expected on the basis of the distinct initial pool and on the key role of randomness and sampling in the first cycles. At generation 24, 13.0 and 11.0 have grown to more than 85% and 53% of DNAi with $\omega = 13$ and $\omega = 11$, respectively. 13.0 and 11.0 likewise have a weak self-interaction (*Figure 3c*), corresponding to hairpins that provide a mild defense to mutual interactions, as schematically sketched in *Figure 4*, panel b1. The hairpins involve most of the bases that are complementary to the resource (yellow shading), leaving however a few unpaired bases that might act as toehold to initiate resource binding. The resource binding takes place at its 3' terminus (panel b2). A bit puzzled about the success of these two species we also explored their capacity of mutual interactions and found that both of them are capable of forming homodimers, as shown in panel b3 for 13.0 (binding energy marked in *Figure 3d*). Such dimers can bind resources at both ends (yellow shading regions), giving 13.0 and 11.0 self-screening and divalent binding capacity. We argue this to be a crucial drive toward success of these two species, which also explains why their growth has such an increment in the latest generations, following the probability of dimer formation.

The third most populous sequence (species 8.0) is characterized by a weak resource interaction which includes a bulge (see *Figure 3c*). While its survival rationale appears clear - good defensive self-binding (panel c1) and interaction targeting the 3' end of the resource (panel c2) - it is hard to accept this could be the cause of the success. Again, inspecting the mutual interaction we found that 8.0 efficiently interacts with 13.0 (as marked in *Figure 3d*), in a region (frame in pink) that does not conflict with the capacity of 13.0 to bind the resource (panel c3). We speculate that this parasitic capacity of 8.0 adds to the weak intrinsic binding to lead to a remarkable success of this species.

As a test to this concept, we focus on species 5.1, whose binding strength to the resource is the weakest in the top 10 species and one of the weakest among all the survivors of generation 24 (*Figure 3c*), a feature that questions its success. However, by inspecting mutual interactions, we find its binding to 11.0 to be strong and stable (pink frame in panel d1 and *Figure 4d1*). An analogous behavior is found for species 5.0 and its interaction to 12.0 (panel d2), confirming that parasitism is an emerging successful survival solution in ADSE.



**Figure 4.** Natural history of DNA individuals (DNAi) species (Oligo1 data). (**a**) Fraction $F$ of {DNAi} that belong to a choice of specific species as a function of the affinity-based DNA synthetic evolution (ADSE) cycles, in linear (left) and logarithmic (right) scale. Arrows connect the initial condition (one individual per species at cycle 0) to the earliest detection via sequencing, across the six orders of magnitudes gap (gray shading). The same growth is assumed for species 12.1, suggesting its appearance by mutation occurred at generation 9. (**b–f**) Self-interactions (**b1, c1, e1**), resource interactions (**b2, c2, e2, d1, d2**), and mutual interactions (**b3, c3, e3, d1, d2, f**) of selected species, sketched as per the NUPACK output. Nucleobases are color coded (G - black, C - blue, A - green, T - red). Paired bases are connected. Double and single stand regions are represented as straight and curved lines, respectively. As in *Figure 1a*, terminal blocks of DNAi are marked as graphic double helices colored according to the legend of panel a, and beads as sketched yellow spheres. Yellow shading: section of DNAi complementary to resources. Pink frames: regions of hybridization between DNAi.

*Figure 4 continued on next page*

*Figure 4 continued*

(**b**) Interactions involving species 13.0 including its homodimerization (**b3**). (**c**) Interactions involving species 8.0, including its binding to 13.0 (**c3**). (**e**) Interactions involving species 12.1, including its homodimerization (**e3**). (**d and f**) DNAi heterodimers interactions suggesting parasitism (**d1**), and possibly mutualism (**d2**) and mutual damage (**f**). The online version of this article includes the following figure supplements.

The online version of this article includes the following figure supplement(s) for figure 4:

**Figure supplement 1.** Evolution of the intra-species interaction strengths ($\langle\Delta G_{pp}\rangle$).

**Figure supplement 2.** 2D probability distribution $p(\omega, x_R)$, as a function of time (panels **a-f**).

We then inspect 12.1 since it has largest $\Delta G_{DR}$ and $\Delta G_{\text{self}}$ among the top 10 species (***Figure 3c***). Species 12.1 has all the features to succeed in ADSE: it forms a weak hairpin with the bases involved in the binding with the resources and a strong hairpin that protects the rest of the ssDNA segment (panel e1); it strongly binds to the resource with the bond ending at its 3' terminal (panel e2); it forms homodimers capable of double resource binding, analogous to that of 13.0 (panel e3). Remarkably, 12.1 emerges only late in ADSE (***Figure 4a***, blue dots), a very unusual behavior when compared to the other many species we have considered. The combination of this late appearance and of the remarkable growth in $F$ afterward suggests that 12.1 is the outcome of one of the rare *mutations* that can occur even with the high-quality PCR that we adopted. By assuming an initial growth analogous to that of dominant species (black arrows in panel a), we argue that such mutation occurred around generation 9, where we have found ancestors having sequence equal to 12.1 except for 4 bases.

$F$ has a monotonic increase for most of the species that become dominant in the latest generations. This behavior is however not at all general. Non-monotonic $F$, growing during in the first generations and decreasing to extinction afterward, is the general behavior for the majority of species that do not become extinct in the first cycle, as also shown - through their average behavior - in ***Figure 2h***. An example of these is species 7.X (empty dots in ***Figure 4a and b***).

More intriguing is when a non-monotonic behavior is observed for one of the dominating species. This is the case of 11.1, whose growth can be again attributed to an effective screening and a good binding to the resources, but whose decrease is harder to justify. To understand it we again investigate mutual interactions, and we find a strong 11.0–11.1 bond (sketched in panel f), which, differently from the DNAi-DNAi complex examined above, competes with resource binding. We thus speculate that the fall of 11.1 is a consequence of the rise of 11.0, which has become frequent enough to make the formation of the complex 11.0–11.1 probable, which lowers the survival probability of both species. This could also explain why species 11.0 has such an irregular growth pattern, with a slowing down when 11.1 becomes highly populated.

The insight gained by these case studies enables appreciating how self and mutual interactions can affect the fate of species in ADSE beyond what we could discern through distributions and average quantifiers. Indeed, this analysis shows that the success of the dominant ASDE species can be achieved through different combinations of resource and mutual DNAi interactions, a fact that makes the evolution of population so varied and generation dependent and explains why, in the latest generations of ADSE, the population distribution $p(\Delta G)$ (***Figure 3a***) is so irregular.

## Discussion

We have introduced here ADSE, a synthetic molecular evolution protocol, that exploits sequencing technology and DNA interaction computability to provide a test-bed for key concepts in ecology and evolution. Although the simple scheme of ADSE enables to perform studies in which mutations and resource drifts can be introduced, we performed experiments by adopting the most simple environment within this protocol, i.e., by holding fixed the capture sequence and minimizing mutations. This choice was aimed at achieving a condition dominated by competition and selection, which enables investigating the nature of fitness in this simplified evolution process.

Our experiments and their comparison with theoretical models indicate that fitness is not a simple function of the direct competitive advantage of strong binding to the resources, as it could have been naively expected. Resource binding is indeed the dominant factor in the first part of the evolution (cycles 1–5), producing a fast growth of $\langle\omega\rangle$ compatible with a strong dependence of the survival probability on $\omega$ ($f \propto \omega^3$). However, as $\langle\omega\rangle$ reaches a value corresponding to bonds of moderate stability

(**Woodside et al., 2006**) - approximately from generation 6 onward - the selective pressure related to resource binding decreases ($f \propto \omega$), a condition that enables appreciation of other factors at play. Which specific factors is suggested by the different growth patterns of $\Delta G_{DR}$ and $\Delta G_{\text{self}}$, the former dominating the first cycles while the latter drives the later stages. However, the quantification of binding energies is not sufficient to predict the fate of individual species, which also depends on interaction details (location of binding and hairpins, secondary structures) and possibly also on kinetics.

In fact, by design, DNAi can interact with themselves, both internally - forming hairpin-like structures - and mutually - leading to the formation of complexes. These interactions can either coexist - when they involve non-overlapping sequences - or they can become competitive. Our experiments reveal that, among the wide variety of conditions made available by the initial random seed, ADSE generally promotes those that combine a good interaction with the resources and a structure capable of screening from mutual DNAi interactions conflicting with binding to the beads. By exploring hybridization-dependent DNAi secondary structures - the only ones accessible with simple analytical tools, we found three basic motifs: hairpins due to self-interactions (as for species 12.1), formation of DNAi homoduplets (as for species 13.0 and 11.0), and formation of dimers of DNAi belonging to distinct species. The latter can bring to distinct prototypical behavior: it can be the basis of *parasitism*, as in the case of the pairs 8.0–13.0 and 5.1–11.0, it can lead to *extinction* with no benefit for either species (as in the case of 15.0–12.0), or it can provide - at least in principle - *mutualism*. While we do not have a solid proof, clues suggest that cooperation might be active in the case of the interaction between species 5.0 and 12.0, since 12.0 has a weak self-defensive system and since the populations of 12.0 and 5.0 evolve very similarly. Finally, we cannot exclude a possible role of the formation of multi-DNAi multivalent complexes. This is the case of species 12.0 and 15.0 whose pattern of multivalent mutual interactions can, in principle, allow higher order interactions (see Appendix 2).

Fitness in ADSE is the outcome of a complex interplay of all these elements. The history of single species in **Figure 4** indicates that the nature of fitness is beyond what can be captured by analyzing the probability distributions of the relevant parameters describing structure and interactions. Moreover, the variety of population evolution patterns enlightens the fundamental fact that despite the constancy of resources, the progressive modification of the ecosystem brings about a change in the competition modes, and thus in fitness. A forthcoming work focused on a limited number of species will be devoted to better disentangle the nature of fitness in ADSE.

The evolution of the ADSE ecosystem, which corresponds to a marked decrement of its entropy - with 2/3 of the initial species become extinct in the first five generations - and potentially terminating with the indication of a single winner species leads instead, in the last cycles, to a significant number of dominant species all still growing at the expense of subservient ones, indicating that in their direct competition none of them is strongly prevailing despite - or maybe thanks to - their distinct survival strategies and their very different population share. Hence, while the niche hypothesis could still be verified in the long run - beyond the experimental limitations due to PCR - it is clear that its drive is weak, suggesting that it could be overcome by environmental fluctuations, thus allowing for coexistence of different species even in a single niche environment.

## Materials and methods

Our experimental design takes advantage of a selective capture mechanism where magnetic beads carrying ssDNA filaments of fixed length and sequence (resources) target DNAi (**Figure 1—figure supplement 1**) present in a DNA library based on their level of complementarity. This process of selection is carried out through subsequent steps that are described in detail in the next paragraphs and represented in **Figure 1—figure supplement 2**.

### Library design

The DNA library contains 100-nt-long sequences where a randomized central region of 50 nucleotides is flanked by 25-nt-long fixed sequences at both its 5' and 3' ends. The fixed regions provide an anchor point for primer annealing, required to perform PCRs during the amplification phase (see below). These terminal segments are made inactive by hybridization with oligomers of perfect complementarity (blockers), so that they are not involved in the selection phase. The blockers, as well as the fixed regions of the DNA library, have been designed to avoid hybridization with the resources.

In addition, the blockers carry a phosphate group at their 3′ end to prevent them from functioning as primers during the PCR amplification. Following the above described criteria, we designed two sets of sequences (see list of sequences in the additional files), called Oligo1 (whose results are presented in the main text) and Oligo2 that was used as a replication experiment. All the oligonucleotides used in this work were purchased from Integrated DNA Technologies, Coralville, IA, USA.

### Beads preparation

The capture of DNAi within the DNA library was performed with carboxylic acid magnetic beads (M-270 Dynabeads, Invitrogen, Carlsbad, CA, USA) coated with the resources. The resources are 20-nucleotide-long, 5′-amino-modified oligonucleotides. Their coupling to the beads surface was performed according to the manufacturer's instructions. Following the activation and coupling procedure, the beads were washed in Tris-HCl (50 mM, pH 7.4), and stored in the same buffer in single-use aliquots.

### Sample preparation

The starting samples were prepared in 1× SSC buffer (0.15 M sodium chloride, 15 mM sodium citrate, pH 7.0). In detail, the Oligo1 (or Oligo2) library (0.75 nmol) was mixed with the blockers (2.25 nmol) in 1:3 molar ratio to saturate all the available interaction sites between the blockers and the fixed regions of the DNA library. The sample was then denatured at 95°C for 5 min, then slowly brought to room temperature using a thermal cycler (MasterCycler Nexus Gradient, Eppendorf, Hamburg, Germany).

### Selection phase

The beads, coupled with the resources, are mixed with the sample, prepared as described above. The capture of the DNAi is carried out at 40°C for 2 hr in stirring (600 rpm, ThermoMixer, Eppendorf). Once the selection phase is completed, the sample is incubated for 2 min on a magnet and the supernatant is removed. The beads, that are now bound to the captured DNAi, are washed three times in SSC 1× buffer to eliminate the aspecific sequences. Finally, the beads are resuspended in water and incubated for 5 min at 60°C to recover the DNAi from the resources. The sample is quickly placed on the magnet and after 2 min the supernatant containing the selected DNAi is collected and then quantified by NanoDrop (Thermo Fisher Scientific).

Control experiments were performed showing that no artifact in ADSE are introduced in ADSE because of non-specific interactions with the magnetic beads (see Appendix 1, Experimental controls).

### Amplification phase

The captured DNAi are then amplified by PCR with Q5 Hot Start High-Fidelity DNA Polymerase (New England Biolabs, Ipswich, MA, USA). DNA samples were diluted 10 times and 3 μl were used for each PCR. The PCR was performed in a final volume of 25 μl using 0.25 μl of polymerase, and each primer had a final primer concentration of 2.5 μM. To allow the regeneration of the single-strand library, the reverse primer was designed to carry a ′ biotin modification (see next paragraph for details). The thermal protocol was the following: (i) denaturation step at 98°C for 2 min, (ii) 28 cycles characterized by three thermal steps: 10 s at 98°C, 10 s at 68°C for Oligo1 (69°C for Oligo2) and 1 s at 72°C, (iii) 2 min at 72°C. The annealing temperature was kept higher than conventional protocols to ensure that hairpins or DNAi dimers are melted. For each generation, 10 different PCRs were performed. The PCR products were then checked for size on a 2% TBE 1× agarose gel.

Control experiments were performed showing that no artifacts are introduced in ADSE from the amplification steps (see Appendix 1, Experimental controls).

### Regeneration phase

PCR products were purified by precipitation with 2.5 volumes of ethanol and 0.1 volumes of sodium acetate 2 M, pH 5.2. After an overnight precipitation at –20°C, the pellet was washed with ethanol 75%, resuspended in water and quantified using the Qubit fluorometer (Invitrogen, Waltham, MA, USA). The single-strand regeneration was performed with streptavidin-coated magnetic beads (M-270 Dynabeads, Invitrogen, Carlsbad, CA, USA) according to the manufacturer's instructions. Briefly, the PCR product was incubated for 15 min on a rotator with the magnetic beads. After two washes with binding and washing buffer (10 mM Tris-HCl pH 7.5, 1 mM EDTA, 0.2 M NaCl) and a third one with

SSC 1× buffer, an alkaline denaturation was performed by incubating the beads with 150 mM NaOH for 10 min to induce the separation of the two DNA strands. This way it is possible to recover the unlabeled DNA strand in solution. The ssDNA is then collected and buffered with 1.25 M acetic acid and TE 1×. The regenerated ssDNA is ready for the next round of selection and amplification.

## Sequencing of the recovered products

To check the growth of DNAi species in our experimental model, some generations were sequenced through next-generation sequencing techniques. To prepare the sample for sequencing, the DNAi species captured as described in the 'Selection phase' paragraph were first PCR-amplified using the same conditions discussed before. However, in this case, both forward and reverse primers were unlabeled. After PCR purification, performed as already described (see 'Regeneration phase' paragraph), the products were quantified and used to obtain libraries. About 300 ng of DNA were used as starting material for the NEBNext Ultra II DNA Library Prep Kit for Illumina (New England Biolabs), and libraries were prepared following the manufacturer's instruction. Sequencing was performed with the NextSeq 550 sequencer (Illumina, San Diego, CA, USA) and a paired-end strategy to obtain 75-nt-long reads.

A technical replica was performed to assess the fluctuations intrinsic to sequencing in the context of ADSE (see Appendix 1, Experimental controls).

## Replicas

The experimental workflow was repeated on two different sets of DNA libraries: Oligo1, whose results are described in the main text, and Oligo2, used as a replicate. The experimental conditions and procedures were the same for both sets of oligonucleotides, the only differences being the sequences of the fixed parts of the DNA libraries (and hence of the blockers and primers) and of the resources (see list of sequences in the additional files). As a consequence, also the PCR conditions had to be adjusted, in terms of the annealing temperature that is one degree higher for Oligo2.

## Eco-evolutionary algorithm

To support experimental observation with a simple abstract model, we developed an evolutionary algorithm where a population of $N_p$ sequences evolves in presence of $N_r < N_p$ shorter sequences. In particular, $N_r$ individuals in this population are selected at each cycle depending on their fitness, i.e., on their affinity to the resource, expressed via their $\omega$ with it. These survived sequences are then amplified by a factor of (roughly) $N_p/N_r$. In this work, two types of fitness functions have been explored: the first one is merely linearly proportional to the $\omega$ of each individual (i.e. $\frac{\omega}{\Sigma\omega}$, where $\Sigma\omega$ is the sum of the $\omega$ values of the whole population), while the other one is a modification of the previous fitness which sets it to be $\frac{\omega_{th}}{\Sigma\omega}$ beyond a threshold value $\omega_{th}$. The code is written in C++, exploiting MPI *Message Passing Interface Forum, 2021* and Armadillo (*Sanderson and Curtin, 2016*; *Sanderson and Curtin, 2018*) libraries for acceleration.

## NUPACK calculations

We resorted to NUPACK for nucleotide sequences analysis, for the prediction of their free energies at equilibrium, either alone or when binding to one or two other oligomers. Schemes like those shown in *Figure 4* have been obtained via the NUPACK web application (*Zadeh et al., 2011*), while massive $\Delta G$ calculations have been performed with custom Python codes exploiting NUPACK Python package (v4.0.0.27) (*Fornace et al., 2020*; *Dirks et al., 2007*). The model is dna04 and ensemble='some-nupack3'; T=40°C and $[Na^+] = 0.24$ M, as in the experiments. Concentration of each species in the tube has been arbitrarily set to $10^{-6}$ M. Detection of hairpins and other secondary structures has been performed visually (thanks to the oxView software; *Bohlin et al., 2022*; *Poppleton et al., 2020*).

## Acknowledgements

TB acknowledges support from MIUR-PRIN (Grant No. 2017Z55KCW). FM and SS acknowledge CloudVeneto (http://cloudveneto.it, *Andreetto et al., 2019*) for the use of computing and storage facilities, through the SEDES and HPC-Physics projects.

## Additional information

### Funding

| Funder | Grant reference number | Author |
| --- | --- | --- |
| MIUR-PRIN | 2017Z55KCW | Tommaso Bellini |
| CloudVeneto | | Samir Suweis |

The funders had no role in study design, data collection and interpretation, or the decision to submit the work for publication.

### Author contributions

Luca Casiraghi, Data curation, Investigation, Visualization, Methodology, Writing – original draft, Writing – review and editing; Francesco Mambretti, Data curation, Software, Formal analysis, Validation, Visualization, Methodology, Writing – original draft, Writing – review and editing; Anna Tovo, Data curation, Software; Elvezia Maria Paraboschi, Data curation, Supervision, Validation, Investigation, Methodology, Writing – original draft, Writing – review and editing; Samir Suweis, Conceptualization, Data curation, Formal analysis, Supervision, Validation, Methodology, Writing – original draft, Writing – review and editing; Tommaso Bellini, Conceptualization, Supervision, Validation, Investigation, Methodology, Writing – original draft, Writing – review and editing

### Author ORCIDs

Francesco Mambretti (iD) http://orcid.org/0000-0002-3712-3595
Tommaso Bellini (iD) http://orcid.org/0000-0003-4898-4400

Reviewer #1 (Public Review): https://doi.org/10.7554/eLife.90156.3.sa1
Reviewer #2 (Public Review): https://doi.org/10.7554/eLife.90156.3.sa2
Author Response https://doi.org/10.7554/eLife.90156.3.sa3

---

## Additional files

### Supplementary files

• Supplementary file 1. content. Sequence list 1: Sequences of the oligonucleotides used in this work for the Oligo1 and Oligo2 datasets. Sequence list 2: Oligo1 cycle 24 top 10 sequences. Sequence list 3: Oligo1 other relevant sequences.

• MDAR checklist

### Data availability

Datasets from DNAi sequencing are available in FASTQ format at https://doi.org/10.5061/dryad.5tb2rbpbs. The Jupyter Notebooks used for the present analysis are available at https://github.com/francescomambretti/stat_phys_synthetic_biodiversity, (copy archived at *Mambretti, 2024*).

The following dataset was generated:

| Author(s) | Year | Dataset title | Dataset URL | Database and Identifier |
| --- | --- | --- | --- | --- |
| Mambretti F, Casiraghi L, Tovo A, Paraboschi EM, Suweis S, Bellini T | 2024 | Synthetic Eco-Evolutionary Dynamics in Simple Molecular Environment | https://doi.org/10.5061/dryad.5tb2rbpbs | Dryad Digital Repository, 10.5061/dryad.5tb2rbpbs |

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

## Appendix 1

### Data analysis protocol

Since we performed a paired-end sequencing, for each experimental cycle two FASTQ files were generated (R1 and R2).

The data shown in the paper all come from the cumulative analysis of R1 and R2 files. To analyze such files, we have developed an ad hoc package (in Python and C++), currently maintained at https://github.com/francescomambretti/stat_phys_synthetic_biodiversity, (copy archived at *Mambretti, 2024*). Here follows a scheme of the main steps of the analysis

1. Read FASTQ file, extracting the list of ssDNA sequences. Reverse sequences are reverted and complemented, so to have all the sequences oriented as $5' \to 3'$. Fixed sequences at the ends are removed in this way: if the full blocker is present, then the 50 bases after it are retained. If some of the last bases of the blocker are lost (it may happen), then just remove the primer and keep the subsequent 50 bases. Slightly different choices for this cut of the sequences turned out to be substantially irrelevant for all the results.
2. Each sequence can be either saved or discarded, according to some filtering procedure. Criteria include the minimum quality of the read ($\geq 10$, measured according to the *Phred* score), the minimum and maximum allowed length, the presence of invalid nucleotides (indicated by 'N' in the FASTQ file) and other checks (see below in particular for PCR by-products).
3. The 'survived' sequences are processed, computing the related observables. Unique sequences are identified and their degeneracy is computed. This allows to monitor the time evolution of the abundance of each species (where the equivalence: DNA sequence-species holds).
4. For each of these unique ssDNAs, we compute the indicators quantifying the 'affinity' with the target sequence: in particular, the MCO and eventually the binding free energy with the resource and/or other oligomers.
5. Several post-processing options are available, such as: generating histograms of the ssDNA population based on their affinity with the resource; tracking the time evolution of dominant strands; determining the fraction of the total population covered by the $n$ most abundant strands. All these quantities can be plotted by suitable Python scripts, provided in the GitHub repository.

### By-product formation from PCR amplification

During the amplification phase, described in our workflow, we observed the formation of PCR by-products. This phenomenon is known in SELEX systems, especially when an oligonucleotide library characterized by high heterogeneity is PCR-amplified (*Tolle et al., 2014*). The generation of these by-products is caused by unspecific hybridization of the primers and/or of oligonucleotides, leading to PCR products abnormal in terms of length and distribution of random/fixed sequences. To limit this phenomenon we adopted blocker oligos with a phosphate group at 3', so that they do not act as primers, and a stringent PCR protocol, characterized by: (i) a high annealing temperature, (ii) fewer PCR cycles, (iii) a high primer concentration, (iv) a very short elongation step (1 s). Moreover, during the data analysis step, we developed an algorithm to identify and remove these spurious sequences.

PCR by-products are identified among the sequenced oligomers according to the following criteria:

- if either the sequence (after the cut of the blocker as previously described) contains at least $n$ consecutive bases identical to $n$ of the 5' blocker (i.e. an $n$-long MCO with a subsequence of the Primer1-F)
- or it has at least $n$ consecutive bases identical to $n$ of the 3' blocker

then this sequence is considered as a by-product and excluded from the analysis.

*Figure 2—figure supplement 1a* shows how such unwanted sequences become $\approx 30\%$ of the total sampled population at cycle 15 and then the striking majority ($> 95\%$) in the last cycle, setting de facto an upper limit to the number of feasible experimental cycles. On the other hand, as we can see from

*Figure 2—figure supplement 1b* the behavior of $\langle\omega\rangle$ for the by-products is very similar to the one observed considering only regular oligomers, therefore their presence does not affect the main outcome of the experiment.

In *Figure 2—figure supplement 2* the $P(\omega)$ for the first and last cycle of only the by-products of Oligo1 are displayed and compared with the null model prediction. The main evolutive trend is also present in these sequences, which have however been discarded from all the analyses.

## Intrinsic limitations of the molecular approach

Possible biases in our workflow are intrinsic to both PCR and Illumina sequencing technologies (e.g. sequencing of homopolymers or amplification of GC-rich sequences). However, in our experimental procedure we implemented approaches and solutions to reduce them. Regarding PCR, we developed an optimized thermal protocol, characterized by very short denaturation, annealing and amplification steps performed at high temperatures. Regarding Illumina sequencing, we can't rule out a bias against specific sequences (e.g. homopolymers), which however should not be captured during the selection step, due to the design of the resource. Also, the libraries subjected to sequencing are characterized by a low complexity since, according to the experimental design, the first and last 25 nucleotides are the same for all DNAi. The complexity further decreases during cycles because of the decrease in DNAi diversity. This could constitute a problem, since in the design of Illumina instruments nucleotide diversity, especially in the first sequencing cycles, is critical for cluster filtering, optimal run performance, and high-quality data generation. To overcome this limitation, the obtained libraries were run together with much larger and more complex and diverse library preparations (and as a consequence the number of reads we obtained was in the range of a few millions per run, a small fraction of the total).

## Experimental controls

To check the specificity and the robustness of our approach, we implemented different control steps: (i) a negative control with bare beads, (ii) a technical replicate of the sequencing, and (iii) a PCR replica.

Negative control. First, we tested the effect of a possible non-specific binding of DNAi to the beads on the selection process. Beads were prepared as described in the Materials and methods section, but they were not coated with the resources. The selection step was performed following the described protocol, and six selection cycles were carried out. We were able to recover DNAi after each selection steps, due to an non-specific binding between the beads surface and DNAi. After sequencing cycle 6, we compared the results with the corresponding cycles of the Oligo1 and Oligo2 experiments. The most abundant sequence in the negative control had a relative occurrence of 0.05%, whereas the dominant strand in the sixth generation in Oligo1 and Oligo2 had an abundance of 8% and 16%, respectively, i.e., 40–80 times larger. This indicates that the natural drift provoked by non-specific interactions with bare beads is more than one order of magnitude smaller than the selection induced by the affinity with the resource.

Technical replicate. Second, we tested the effect of sampling on the consistency of the sequencing results. Since only a small fraction of the recovered sample is actually analyzed, we sequenced twice the library derived from Oligo1 cycle 9, and we identified in both replicates the most abundant DNA species, defined as those with at least 100 sequencing reads (corresponding to 27.42% of the total reads). Among the 800 DNA species that satisfied this requirement, 93.6% are found in both replicates; the remaining ones are characterized by a low representation, close to 100 reads. Subsequently, we compared the abundance of DNAi populations between the two replicates (*Figure 2—figure supplement 3*), by calculating the ratio between the number of reads of each shared DNA species in each replicate, after normalizing for the total number of reads obtained from sequencing. We found an average ratio of 0.965 (standard deviation = 0.119), and we observed significant fluctuations only for the least populated species. Overall, these results indicate that the effect of sampling (and thus the sequencing readout, as well as possible PCR amplification errors) don't have a significant impact on the selection process.

PCR effects on DNAi selection. Third, we tested the effect of PCR amplification on the selection process. Starting from Oligo1 cycle 9 sample, we performed an additional PCR amplification round (equal to those used to connect ADSE generations) and we proceeded directly to sequencing with no beads-selection step. We then compared the ensemble of oligos obtained in this way, which

we named Oligo1 'cycle 9 replica', with both the original Oligo1 cycle 9, and with Oligo1 cycle 10. We sampled 20 times $4 \times 10^5$ sequences from the cycle 9 dataset, from cycle 9 replica and from cycle 10 with a bootstrap approach. To compare the three systems we extracted the fraction of the population of each covered by the 10 most abundant individuals (*Figure 2—figure supplement 4*). We found that the 10 most abundant species represent on average the 8.8% ± 0.1% of all sequences in Oligo1 cycle 9 and the 8.5% ± 0.1% in its replica, thus indicating a statistical compatibility within three standard deviations (calculated on the 20 subsamples). Notably, across the 20 subsampling, for each of the analyzed condition, the 10 most abundant sequences are almost always the same. In particular, the first 8/9 are always the same, possibly shuffled. This result should be compared with the top 10 fraction in cycle 10, which is 13.6% ± 0.1%, far beyond the statistical compatibility with cycle 9, with cycle 9 replica and with the distance between the two.

## Appendix 2

### Further analyses of the eco-evolutionary model

We here show some further results and sensitivity analyses of the eco-evolutionary stochastic individual-based model. In *Figure 2—figure supplement 7* we show how the eco-evolutionary model can describe the emergent order and complexity, that is a loss of the initial huge diversity of the random strings toward few dominating species, during evolution. This behavior can be quantified (i) by the ratio CR between the zipped file size (*Benedetto et al., 2002*) and the original file size of all the sequences oligomers in the ecosystem or (ii) by computing the Shannon entropy of the relative species abundance (*Hill, 1973*). We can see that the model is able to reproduce, at least qualitatively, the marked decrease of diversity observed during evolution.

In *Figure 2—figure supplement 8a* we show the outcome of the model if we do not include the saturation at $\omega_{th} = 10$. As we can see, the effect is evident as after a few cycles the experimental $\langle \omega \rangle$ lies significantly under the model (which promotes too much the strongest sequences). Similarly, *Figure 2—figure supplement 8b* shows the evolution for the average MCO in the case we use only $\gamma = 1$ in the simulation (here, $\omega_{sat} = 11$). It is very apparent how in this case the simulated initial evolution is much slower with respect to the observed from the data, while the final trend is captured. These results indicate that both saturation and the non-linear fitness time-dependent fitness function are fundamental ingredients to describe the empirical evolution of $\langle \omega \rangle$. Note that in both cases of the above picture, there is a 10% random sampling drift.

Finally, *Figure 2—figure supplement 9a* and *Figure 2—figure supplement 9b* show the simulation results for different total number of individuals ($N = 10^6$, $10^5$, and $10^4$) and resources ($R = N/100$, keeping this ratio fixed) and for different initial conditions (i.e. different sequences populations), respectively. We can see that the qualitative results concerning $\langle \omega \rangle$ as a function of time are robust, beyond a given system's size (the silver data in *Figure 2—figure supplement 9a* are already of a system's size large enough, while the blue ones correspond to a too small system). Considering a different initial (random) population of sequences, keeping fixed all the other hyperparameters (in particular $N = 10^6$ and $R = 1064$), as shown by *Figure 2—figure supplement 9b* preserves the qualitative agreement with the experimental data, assessing the substantial independence on the initial conditions. Overall, these further analyses show the robustness of the qualitative behavior of results of the eco-evolutionary model, pointing out to two relevant phases of the evolution of the oligomers, as highlighted in the main text.

### Further results on the interactions between oligomers

In this section we highlight some further results on the interactions between species and resources and among oligomers, respectively.

*Figure 4—figure supplement 1* shows the average pair-wise interactions between 10 random strings (as representative of the whole population) during the different evolutionary cycles ($t = 0, 9, 18, 24$), $\langle \Delta G_{pp} \rangle$. Such averages have been computed by giving to NUPACK a pool of 1000 sequences, randomly subdivided into 100 groups of 10 sequences each. We can see that, during evolution, the selected oligomers increase their average interaction strength, a signature that the evolution favors not only strings that have higher overlap with the resource, but also that interact in some way with the other species.

*Figure 4—figure supplement 2* is the 2D probability distribution $p(\omega, x_R)$, computed from experimental data, to simultaneously have a given MCO $= \omega$ and $x_R$, the rightmost basis of the resource involved in the formation of such an MCO. Noticeably, we start from a distribution peaked in $\omega = 4$ and $10 \leq x_R \leq 12$, essentially corresponding to purely random binding. Across cycles, the probability distribution is deeply modified, with a shift toward the bottom right corner. Along the horizontal axis this simply reflects the shift in the average $\langle \omega \rangle$ of the population, and also the fact that winners have large overlaps. Interestingly, we observe a significant shift of $x_R$ toward high values, meaning the MCO are preferably formed far away from the bead surface, near the free end of target strands.

### Results of the experimental replica

In *Figure 2—figure supplement 5* and *Figure 2—figure supplement 6* show the main outcomes of the replica experiment Oligo2. The evolution of the average MCO $\langle \omega \rangle$ over cycles is qualitatively

the same of that one found in Oligo1: we have a initial steep growth $\langle \omega \rangle$, compatible with a non-linear fitness, and then we observe a slower increase for the last cycles, saturating for $\langle \omega \rangle \approx 7$. The histogram shows the whole $P(\omega)$ at cycles $t = 0$ and $t = 18$. As for what we found in Oligo1, at $t = 0$ the behavior of $P(\omega)$ is compatible with our null random model, while at $t = 18$ the oligomers with higher $\omega$ are selected, and the tail of $P(\omega)$ is larger than the one expected by chance.

These results show that indeed the main outcome of Oligo1 are replicated by Oligo2.

# Appendix 3

## A combinatorial null model with no evolution

As a first step to understand what happens in our experiments, we have built a null model, assuming purely random attachment sites, i.e., there is not any search for the relative positions of binding filaments so to maximize the MCO. In this way we have a null expectation for the typical MCO distribution in case of oligomers insensitive to the presence of target sequences.

Let us call $a$ the number of consecutive bonded pairs between oligomers 1 (e.g. the species) and 2 (e.g. the resource), with $\max_{r_{1,2}} MCO = \omega$ and $R_{1,2}$ being the relative position of the left ends of the two filaments, as also explained in **Message Passing Interface Forum, 2021**, and in the main text. Consider the initial population being uniformly random selected (in the high $4^L$ dimensional space of all possible DNA strands). The first quantity we are interested to calculate is $\pi(a)$, the *probability density of $a$* at time $t = 0$, i.e., before the evolution starts. This distribution is also the distribution of the sub-sample (i.e. the $\approx 10^6$ strands which we are able to sequence at each cycle), if we assume purely random attachment (no competition among the DNAi, no optimization of the attachment site).

The relative position of the two strands is important since the target can be either 'inside' (**Figure 3—figure supplement 1**, panel a) or surpassing one of the borders (**Figure 3—figure supplement 1**, panels b and c) of the longer oligomer after the attachment. In the picture, $R_{1,2} = 0$ when the first nucleotide of the target resource overlaps with the first one of the DNAi species. From now on, we will refer to $R_{1,2}$ simply as $R$.

Our goal consists in finding the MCO of $L$-long strand (the species) with a $l$-long target ssDNA (the resource), where $l < L$. In the following calculations, we therefore assume that $R$ is a random uniform variable ($-l + 1 \leq r \leq L - 1$) and we will calculate $a|_R$ (for this $R$).

Such a problem is sketched in **Figure 3—figure supplement 2** (where $R = 2$, but the following holds for any $R$ in the allowed range), where we indicate with $d$ the number of mismatching bases between the target resource (red) and the DNAi species (blue), with $u$ the number of matching ones, being $d + u := l$. Labeling as $x_i$ the number of consecutive matching nucleotides comprised between two consecutive mismatching bases, as shown in the picture, the problem can be reformulated in terms of counting how many integer solutions can be found for the equation:

$$\sum_{i=0}^{d} x_i = u \tag{1}$$

that corresponds to determine the number of ways in which a set of $x_i$ solving **Equation 1** can be obtained. Note that $x_i$ vanishes between two neighboring pairs of mismatching sites, as marked by the arrows in the picture. In the case reported in the picture, e.g., $x_i = (1, 0, 1, 4, 0, 4, 0, 0, 2)$ and $\sum_{i=0}^{8} x_i = 12$, $u = 12$ and $d = 8$.

**Charalambides, 2008**, provides us with a formula to answer this question. We stress that finding the number of solutions $A_{d,u}$ for the above equation is equivalent to obtaining the probability distribution $\pi(a)$ for the relative occurrence of each $a$ in a sample of DNA strands, with uniformly random extracted relative positions of the two sequences.

Theorem 1.

Let us have the equation $\sum_{i=1}^{n} x_i = u$. Let us suppose that $s_i \leq x_i \leq m_i, \forall i = 1, \ldots, n$ for given integers $s_i, m_i, i = 1, \ldots, n$ with $s \leq u \leq m$, being $s := \sum_{i=1}^{n} s_i$, and $m = \sum_{i=1}^{n} m_i$. Let us define $w_i = m_i - s_i \geq 0 \, \forall i$. The number of integer solutions, having the shape $(r_1, \ldots, r_n)$, of this equation, with the restrictions aforementioned, is given by:

$$A_{n,u}(w_1, \ldots, w_n) = \binom{n + u - s - 1}{n - 1} + \\ \sum_{r=1}^{n} (-1)^r \sum_{\{i_1, i_2, \ldots, i_r\}} \binom{n + u - s - (\sum_{j=i_1}^{i_r} w_j) - r - 1}{n - 1} \tag{2}$$

where the inner sum is performed over all the r-combinations (having the shape $\{i_1, i_2, \ldots, i_r\}$) of the $n$ indices $1, \ldots, n$.

In our case, the resource string may not go beyond the DNAi species boundaries (as the latter is longer than the former). But we can modify the name of some variables to adapt the theorem to our problem and make some constraints explicit:

- From $i = 1, \ldots, n$ it follows that $n = d + 1$ because we have $d + 1$ terms (from 0 to $d$).
- $s_i = 0 \,\forall i$, because each $x_i \geq 0$ by definition. Therefore, $s = 0$.
- At least one of the $x_i$ must be equal to $a$.
- In principle, $m_i$ is equal to $a$ for each $i$ (which means that there are no consecutive overlaps larger than $a$).
- It must be that $a \leq u \leq \min((d+1)a, l)$ (i.e., the total overlap $u$ - which, at most, is equal to $d + 1$ times $a$ - must not be larger than $l$).

In fact, by using $s = 0$ and $d + u = l$, **Equation 2** can be rewritten as:

$$A_{d+1,u}(w_1, \ldots, w_n) = \binom{l}{d} + \sum_{r=1}^{d+1}(-1)^r \sum_{\{i_1,i_2,\ldots,i_r\}} \binom{l - (\sum_{j=i_0}^{i_r} w_j) - r}{d}$$

where the inner sum can be replaced by $\binom{d+1}{r}$ (how many ways are there to dispose the $d + 1$ integer indexes to form $r$ groups):

$$A_{d+1,u}(w_1, \ldots, w_n) = \binom{l}{d} + \sum_{r=1}^{d+1}(-1)^r \binom{d+1}{r} \binom{l - (\sum_{j=i_0}^{i_r} w_j) - r}{d}$$

First of all, $\binom{l}{d}$ can be placed inside the summation because it corresponds to the $r = 0$ case. Now, we have to apply the constraint that at least one of the $w_j = a$, which means having the first term where the $w_j$ are always equal to $a$, minus the second term which includes all the cases where all the $w_j < a$:

$$A_{d+1,u}(w_1, \ldots, w_n) = \sum_{r=0}^{d+1}(-1)^r \binom{d+1}{r} \binom{l-ra-r}{d} -$$
$$\sum_{r=0}^{d+1}(-1)^r \binom{d+1}{r} \binom{l-r(a-1)-r}{d} =$$
$$\sum_{r=0}^{d+1}(-1)^r \binom{d+1}{r} \binom{l-r(a+1)}{d} - \sum_{r=0}^{d+1}(-1)^r \binom{d+1}{r} \binom{l-ra}{d}$$

Note that the previous expressions concern only the cases where $0 \leq R \leq L - l$. However, this should not matter for practical purposes because $R$ does not appear in the above expressions.

Similar formulas can be found when the target surpasses the left/right border: if $> -l + 1 \leq R \leq -1$

$$A_{d+1,u}(w_1, \ldots, w_n)^{(II)} = \sum_{r=0}^{d+1}(-1)^r \binom{d+1}{r} \binom{l + R - r(a+1)}{d} -$$
$$\sum_{r=0}^{d+1}(-1)^r \binom{d+1}{r} \binom{l + R - ra}{d}$$

while if $> L - l \leq R \leq L - 1$

$$A_{d+1,u}(w_1, \ldots, w_n)^{(III)} = \sum_{r=0}^{d+1}(-1)^r \binom{d+1}{r} \binom{L - R - r(a+1)}{d} -$$
$$\sum_{r=0}^{d+1}(-1)^r \binom{d+1}{r} \binom{L - R - ra}{d}$$

Using all the previous results, the general equation for the number of ways which lead to a MCO $a$ is given by:

$$n(a, R) = \sum_{k=1}^{3} n_k(a, R) \tag{3}$$

where the three options are originated by the three options for the relative position of the target and of the DNAi sketched in **Figure 3—figure supplement 1**.

Let us now derive $n_k, \forall k$. In particular, if the target resource is completely within the DNAi species boundaries, then we have

$$n_1(a, R) = n_1(a) = (L - l + 1)4^{L-l} \sum_{d=0}^{l} 3^d n_{1,d}(a), \tag{4}$$

with $(L - l + 1)$ is the total number of possible positions $R$ where the DNAi can attach to the resource, $4^L$ is the number of ways to build a $L$-long string (i.e. all possible species), $3^d$ represents the number of arrangements for the non-matching bases (3 is because we exclude the only matching base) and $n_{1,d}(a) = A_{d+1,u}^{(I)}$ for $0 \leq R \leq L - l$.

Analogously,

$$n_2(a, R) = \sum_{R=-l+1}^{-1} 4^{L-l-R} \sum_{d=0}^{l+R} 3^d n_{2,d}(a, R), \tag{5}$$

where the summations run over the admitted integer $R$ values as well as over the allowed $d$ values. In this case, $4^{L-l-R}$ is the number of ways in which the nucleotides not involved in the attachment can be obtained. Here, $n_{2,d}(a, R) = A_{d+1,u}^{(II)}$. A similar equation holds for $n_3(a)$, where

$$\sum_{R=L-l+1}^{L-1} 4^R \sum_{d=0}^{L-R} 3^d n_{3,d}(a, R)$$

and $n_{3,d}(a, R) = A_{d+1,u}^{(III)}$.

To get the distribution of the overlap measure $\pi(a)$, it suffices to divide $n(a, R)$ by the total number of ways in which we can build a string, i.e., $4^L$ times the number of possible positions $R$ where the attachment can happen, i.e., $L + l - 1$.

$$\pi(a) := \frac{n(a, R)}{\sum_{j=0}^{l} n(j, R)} = \frac{n(a, R)}{4^L (L + l - 1)} \tag{6}$$

$\pi(a)$ is exactly the probability density to find a given MCO $= a$ between a $l$-long and a $L$-long sequence which found themselves attached in a random position $R$.

The underlying probability distribution $\pi(a)$ is naturally discretized and can be represented as an histogram, with the occupancy probability for each bin $a$ (the affinity to the resource). $\pi(a)$ is represented in **Figure 3—figure supplement 3a** as orange bars; gray bars in the same panel represent the average of 50 independent simulations with $K = 10^6$ 50mers uniformly generated, where their overlap with the target strand - once uniformly random extracted their relative position - is measured.

The analytical result for the null model and simulations yield almost exactly the same results, with the difference that for large MCO we have strong sampling effect in the simulations. In order to understand the possible biases in the initial frequency of each nucleotide in the starting population, we run $N = 50$ independent simulations with a non-uniform frequency for A,C,G,T in the sequences. For instance, the panel b of **Figure 3—figure supplement 3** compares the $\pi(a)$ obtained with A,C,G,T having a probability of 25% (gray data) with the corresponding distribution obtained having: A=23%, C=19%, G=29%, T=29%. The average bin occupancy does not change significantly, but only the large $a$ tails are slightly affected by the bias and the error bars are much larger for the majority of the $a$ values.

We furthermore set a threshold $T$ to model the physical constraint that if $a < T$, the two ssDNAs cannot bind. We thus obtain a null model physically constrained distribution $\pi'(a)$, where $\pi'(a) = 0$ if $a < T$, while is the same as before (after a proper normalization) for $a \geq T$. The new analytical distribution $\pi'(a)$ reads as:

$$\pi'(a) = \frac{\pi(a) \times \frac{1}{2}[\text{sgn}(a - T) + 1]}{\sum_{j=T}^{l} \pi(j)}, \tag{7}$$

where $\text{sgn}(0) := 0$. The corresponding pseudo-algorithm to generate random number from such distribution is

- sample a random number $z$ from $\pi(a)$

- if $z < T$, sample another number from $\pi(a)$
- else, save it in the new distribution $\pi'(a)$.

The comparison between analytical and numerical results are reported in **Figure 3—figure supplement 4a** and **Figure 3—figure supplement 4b** (linear and logarithmic scale on $y$-axis, respectively). The cyan bars correspond to the analytical distribution for the null model with physical constraint ($\pi'(a)$), whereas the purple bars represented the average over $N = 200$ independent simulations of $\pi'(a)$ performed via Metropolis Monte Carlo accept-reject technique (**Metropolis et al., 1953**). A custom Python script which simulates this process is available at the GitHub repository associated to this work. The null model that we present in the main text is the physical constraint one, i.e., $\pi'(a)$.

