## [Editor Report · eLife assessment]

In this **important** study, the authors develop a promising experimental approach to a central question in ecology: What are the contributions of resource use and interactions in the shaping of an ecosystem? For this, they develop a synthetic ecosystem set-up, a variant of SELEX that allows very detailed control over ecological variables. The evidence is **convincing**, and the work should be of broad interest to the ecology community, leading to further quantitative studies.

---

## [Referee Report · Reviewer #1 (Public Review)]

This work describes a new and powerful approach to a central question in ecology: what are the relative contributions of resource utilisation vs interactions between individuals in the shaping of an ecosystem? This approach relies on a very original quantitative experimental set-up whose power lies in its simplicity, allowing an exceptional level of control over ecological parameters and of measurement accuracy.

In this experimental system, the shared resource corresponds to 10^12 copies of a fixed single stranded target DNA molecule to which 10^15 random single stranded DNA molecules (the individuals populating the ecosystem) can bind. The binding process is cycled, with a 1000x-PCR amplification step between successive binding steps. The composition of the population is monitored via high-throughput DNA sequencing. Sequence data analysis describes the change of population diversity over cycles. The results are interpreted using estimated binding interactions of individuals with the target resource, as well as estimated binding interactions between individuals and also self-interactions (that can all be directly predicted as they correspond to DNA-DNA interactions). A simple model provides a framework to account for ecosystem dynamics over cycles. Finally, the trajectory of some individuals with high frequency in late cycles is traced back to the earliest cycles at which they are detected by sequencing. Their propensities to bind the resource, to form hairpins or to form homodimers suggest how different interaction modes shape the composition of the population over cycles.

The authors report a shift from selection for binding to the resource to interactions between individuals and self-interactions over the course of cycles as the main driver of their ecosystem. The outcome of the experiment is far from trivial as the individual-resource binding energy initially determines the relative enrichment of individuals, and then seems to saturate. The richness of the population dynamics observed with this simple system is thus comparable to that found in some natural ecosystems. The findings obtained with this new approach will likely guide the exploration of natural ecosystems in which parameters and observables are much less accessible.

My review focuses mainly on experimental aspects of this work given my own expertise. The introduction exposes very convincingly the scientific context of this work, justifying the need for such an approach to address questions pertaining to ecology. The manuscript describes very clearly and rigorously the experimental set-up. The main strengths of this work are (i) the outstanding originality of the experimental approach and (ii) its simplicity. With this setup, central questions in ecology can be addressed in a quantitative manner, including the possibility to run trajectories in parallel to generalize the findings, as reported here. Technical aspects have been carefully implemented, from the design of random individuals bearing flanking regions for PCR amplification, binding selection and (low error) amplification protocols, and sequencing read-out whose depth is sufficient to capture the relevant dynamics. With this setup one can tune the relative contributions of binding selection vs amplification for instance (to disentangle forces that shape the ecosystem). One can also run cycles with new DNA individuals, designed with arbitrarily chosen resource binding vs self-binding, that are predicted to dominate depending on chosen ecological parameters. These exciting perspectives underlie the strong potential of the new approach described in the current study.

---

## [Referee Report · Reviewer #2 (Public Review)]

Summary:

In this manuscript, the authors introduced ADSE, a SELEX-based protocol to explore the mechanism of emergency of species. They used DNA hybridization (to the bait pool, "resources") as the driving force for selection and quantitatively investigated the factors that may contribute to the survival during generation evolve (progress of SELEX cycle), revealing that besides individual-resource binding, the inter- and intra-individual interactions were also important features along with mutualism and parasitism.

Strengths:

The design of using pure biochemical affinity assay to study eco-evolution is interesting, providing an important viewpoint to partly explain the molecular mechanism of evolution.

---

## [Author Response]

The following is the authors’ response to the original reviews.

**Reviewer #1:**
This work describes a new and powerful approach to a central question in ecology: what are the relative contributions of resource utilisation vs interactions between individuals in the shaping of an ecosystem? This approach relies on a very original quantitative experimental set-up whose power lies in its simplicity, allowing an exceptional level of control over ecological parameters and of measurement accuracy.In this experimental system, the shared resource corresponds to 10^12 copies of a fixed single-stranded target DNA molecule to which 10^15 random single-stranded DNA molecules (the individuals populating the ecosystem) can bind. The binding process is cycled, with a 1000x-PCR amplification step between successive binding steps. The composition of the population is monitored via high-throughput DNA sequencing. Sequence data analysis describes the change in population diversity over cycles. The results are interpreted using estimated binding interactions of individuals with the target resource, as well as estimated binding interactions between individuals and also self-interactions (that can all be directly predicted as they correspond to DNA-DNA interactions). A simple model provides a framework to account for ecosystem dynamics over cycles. Finally, the trajectory of some individuals with high frequency in late cycles is traced back to the earliest cycles at which they are detected by sequencing. Their propensities to bind the resource, to form hairpins, or to form homodimers suggest how different interaction modes shape the composition of the population over cycles.The authors report a shift from selection for binding to the resource to interactions between individuals and self-interactions over the course of cycles as the main drivers of their ecosystem. The outcome of the experiment is far from trivial as the individual resource binding energy initially determines the relative enrichment of individuals, and then seems to saturate. The richness of the population dynamics observed with this simple system is thus comparable to that found in some natural ecosystems. The findings obtained with this new approach will likely guide the exploration of natural ecosystems in which parameters and observables are much less accessible.My review focuses mainly on the experimental aspects of this work given my own expertise. The introduction exposes very convincingly the scientific context of this work, justifying the need for such an approach to address questions pertaining to ecology. The manuscript describes very clearly and rigorously the experimental setup. The main strengths of this work are (i) the outstanding originality of the experimental approach and (ii) its simplicity. With this setup, central questions in ecology can be addressed in a quantitative manner, including the possibility of running trajectories in parallel to generalize the findings, as reported here. Technical aspects have been carefully implemented, from the design of random individuals bearing flanking regions for PCR amplification, binding selection and (low error) amplification protocols, and sequencing read-out whose depth is sufficient to capture the relevant dynamics.:We thank the reviewer for summarizing our work and the main findings in a very clear and effective manner.One missing aspect in the data analysis is the quantification of the effect of PCR amplification steps in shaping the ecosystem (to be modeled if significant). In addition, as it stands the current work does not fully harness the power of the approach. For instance, with this setup, one can tune the relative contributions of binding selection vs amplification for instance (to disentangle forces that shape the ecosystem). One can also run cycles with new DNA individuals, designed with arbitrarily chosen resource binding vs self-binding, that are predicted to dominate depending on chosen ecological parameters. I have three main recommendations to the authors:1. PCR amplification steps (and not only binding selection steps) should be taken into account when interpreting the outcome of experiments.1. More generally, a systematic analysis of the possible modes of propagation of a DNA molecule from one cycle to the next, including those considered as experimental noise, would help with interpreting the results.1. Testing experimentally the predictions from the analysis and the modelling of results would strengthen the case for this approach.

Despite its conceptual simplicity, our approach has indeed a few experimental handles that enable exploring a relevant variety of conditions much beyond those described in this paper, of which we are very aware. These involve selection vs. amplification or set the stage to explore competition, parasitism or cooperation among specific species, as the reviewer points out, but also introduce mutations and explore the kinetics of evolution in static or dynamic environments. Ongoing experiments are considering some of these conditions. We modified the text to mention more explicitly these possibilities, which are now mentioned in p11 lines 376-378 and lines 416-417. The three points raised by the reviewer helped us to further improve and clarify strengths and limitations of our work, as detailed below.

Regarding the first point, here are my suggestions :Run one cycle of just amplification vs 'binding + amplification', or simply increase the number of PCR cycles (and subsample the product) to check whether it impacts the population composition, in particular for sequences with predictions derived from the current analysis.

The point raised by the reviewer is indeed very relevant and not discussed in our manuscript. Prompted by the reviewer’s comment, we performed two new experiments to distinguish resource-binding selection from PCR amplification effects.

First, we performed a negative control experiment in which we performed the “selection step” with bear beads, i.e. beads without with no DNA grafted on them. We then compared the results with the corresponding results of the original experiments on Oligo 1 and 2.

After 6 cycles, the most abundant sequence in the negative dataset has a relative occurrence of 0.05%, whereas the dominant strand in Oligo 1 and Oligo 2 has an abundance of 8% and 16%, respectively, i.e. 40-80 times larger.

This indicates that the drift due to non-specific binding + PCR amplification is at least two orders of magnitude smaller than the selection induced by the affinity with the resource.

This results are now cited in p14 lines 468-470, and described in Appendix 1, Experimental controls.

Second, we tested the effect of PCR amplification on the selection process. We exploited the fact that we have aliquots for each generation of our evolution experiment, which we sampled and saved after PCR and before sequencing. We thus chose a specific generation - specifically generation 9 from Oligo 1 experiments - and performed another PCR round we proceeded directly to sequencing with no beadsselection step. We then compared the ensemble of oligos obtained in this way, which we named Oligo 1 “cycle 9 replica”, with both the original Oligo1 cycle 9, and with Oligo1 cycle 10.

We sampled 20 times 4 x 10^5 sequences from the cycle 9 dataset, from cycle 9 replica and from cycle 10 with a bootstrap approach. To compare the three systems we extracted the fraction of the population of each covered by the 10 most abundant individuals. The results are shown in Figure 2 - Figure Supplement 4. In the figure caption further details on the analysis can be found. The similarity between cycle 9 and cycle 9 replica and the marked difference between cycle 9 replica and cycle 10

indicates that the relevant part of the selection is indeed performed by the resourcebinding mechanism, while drifts induced by PCR play a secondary role.

As a further check, we compared the specific sequences across the 20 samples in cycle 9 and cycle 9 replica datasets and found that the 10 most abundant sequences are almost always the same. In particular, the first 8/9 are always the same, possibly shuffled.

These new pieces of evidence are now cited in p14 lines 483-484 and described in Appendix 1, Experimental controls.

Sequencing read-out includes the same PCR protocol as the one used for amplification steps, so read-out potentially has an effect on the composition of the ecosystem. Again, varying the number of PCR cycles is a direct way to test this.

The PCR amplification involved in the read-out might have a minor effect on the sequencing outcome but not on the composition of the ecosystem. In fact, the sample that undergoes sequencing is taken from the pool at each cycle, and not inserted back into it. Thus, it does not participate in the following selection steps. This is specified in the text at p3 line 104

Could self-interactions (hairpins of homodimers) benefit individuals during amplification steps? The role of self-interactions during binding selection steps could also be tested directly over one cycle (again varying the relative weight of the binding vs amplification to disentangle both).

Our choice of conditions for PCR amplification were thought to minimize effects of this type. PCR amplification is carried out at 68 C, a temperature at which, given the level of self and mutual complementarity in the sequences analyzed in the text, hairpins or homodimers should be melted and thus have no effect. This is specified in the text at p. 14 lines 479-480However, if an effect is present, it gives a disadvantage (rather than an advantage) to self-interacting individuals. For the amplification step we used Q5 Hot Start HighFidelity DNA Polymerase, which does not possess strand displacement activity. Therefore, in theory, if during amplification the polymerase encounters a double strand portion, it stops and synthesizes only a truncated product, which will be then lost during the purification step. In other words, sequences with secondary and/or tertiary structures are less likely to be amplified during the polymerization step. As a consequence, a DNAi that is characterized by this kind of structures, will be negatively selected even in the case of optimal binding to the resource, and will be underrepresented in the pool.

About the second point:Regarding the effect of sampling (sequencing read-out), PCR amplification errors: explicitly check the consistency of observations with the expected outcome, in the methods section (right now these aspects are only briefly mentioned in the main text), which would highlight again the level of control and accuracy of the system.

Hoping to have well interpreted the request, we performed a technical replicate sequencing Oligo 1 cycle 9 again and analyzed the sequences that have at least 100 reads (corresponding to 27.42% of the total reads). We find that among the 800 DNA species that have at least 100 reads, 93.6% are found in both replicates. All the nonoverlapping sequences have very low abundance, close to 100.

Moreover, we compare the population size of each DNA species between the two replicas, after having equalized the database sizes. The results are now cited in p14 lines 509-510, In Appendix 1, Experimental Controls and shown in Figure 2-figure supplement 3, where we plot the ratio of the number of reads in the two replicates for each sequence as a function of the number of reads in one. We found an average of 0.965 with a standard deviation of 0.119. High fluctuations are found in the most rare species, as expected.

We think this evaluation indeed strengthens the solidity of our results.

I have a small concern about target resource accessibility: is there any spacer between the ssDNA and the bead? The methods section does not mention any, and I would expect such a proximity between the target DNA and the bead to yield steric repulsion that impedes interactions with random DNA individuals.

Yes, there is a 12-carbon spacer between the bead and the resource, which was inserted exactly to make the resource more accessible. This information is now available in Table 1 of Supplementary Information detailing the sequences used in the experiment. However, as now described in the text (p8 lines 284-286), we observe that the interaction with the resource is always shifted to the 3', the terminal furthest from the bead, indicating some residual issue of accessibility to the resource sections closest to the bead.

Regardless of the existence of a spacer, binding of random DNA molecules to beads instead of the target DNA constitutes a potential source of noise (described for now as '1-x' in the IBEE model), which can be probed by swapping targets, selecting without target etc.

This issue is addressed by the test with bare beads described above, in which we found little effects, corresponding to small 1*x* value.

Is there any recombination potentially occurring during amplification steps? This could be tested with a set of known molecules amplified over 24 amplification steps in a row (no binding step).

It is possible for recombination to occur during the amplification steps. In Appendix 2, the section "By-Product Formation from PCR Amplification", discusses PCR byproducts as aberrant forms of amplification, such as recombination events. We adopted several strategies to limit by-product formation, such as: (i) use of “blockers” characterized by a phosphate group at 3’ end (thus inhibiting their usage during the amplification and allowing a better control of the reaction conditions over the PCR cycles), (ii) a high annealing temperature (to limit the possibility of a spurious primer annealing to the random region), (iii) fewer PCR cycles, (iv) a high primer concentration, (v) a very short elongation step (all these strategies have been implemented to avoid a possible mispriming event between different DNAi, and the formation of concatemers). However, the formation of by-products is a problem inherent to the technique: in fact, it is a known issue for classical SELEX technology (Tolle et al. 2014), mainly due to the random region within the DNAi. Q5 Hot Start High-Fidelity DNA Polymerase (New England Biolabs, Ipswich, MA, USA) has an error rate of <0.44 x 10-6/base.

In classic SELEX technology, the average number of selection cycles is 10. This limitation is partly due to the increase in PCR by-products. As we can see from Figure 2 Supplementary Figure 1, the percentage of PCR by-products is less than 20% at cycle 12, and then increases dramatically in the following cycles. We are performing a series of experiments with known and limited sequences to verify and better understand the phenomenon for future applications of the SEDES platform. On this issue we decided not to modify the manuscript since we think it is already well discussed in Appendix 1.

And the third point:Perform one cycle (or a few cycles) with random DNA individuals, the most frequent individuals at the end of the current experiment, newly designed individuals with higher binding affinity to the target than currently dominating individuals, newly designed individuals with higher propensity to form hairpins or to form homodimers. Such experimental testing of predictions from the data analysis/modeling, typical of a physics approach, would illustrate the level of understanding one can reach with a simple yet powerful experimental setup.

We perfectly agree that the approach we propose and the set of results we obtained call for further investigations that could strengthen analysis and modeling. The final aim we envisage is the understanding, within this simplified approach, of key evolutionary factors such as fitness. Indeed, becoming able to write an explicit fitness function would be a significant new contribution to the understanding of evolutionary processes, even within the limited settings of the ADSE approach, as discussed in the conclusions of the manuscript.

However, undergoing such an analysis is a long and expensive job, which we have started and will be completed in a not immediate future. For this reason, given the already significant body of results we are presenting here, we prefer to keep this paper confined to the study of the evolution of a random DNAi population and discuss in a future contribution the behavior of smaller designed sets of competing, collaborating or parasitic individuals.

Looking ahead, additional stages of investigations will also include mutations - to investigate the kinetics of speciation, and, in an even further stage, the interplay between evolution kinetics and dynamical mutation of resources.

I have a few smaller points:It would be very useful to provide the expected dynamic range of binding free energies (in terms of DeltaG and omega): what is the maximum binding free energy for the perfect complement?

The NUPACK-computed binding free energy of a 20 basis-long oligomer complementary to the resource (*ω*�=20) is -24.36 Kcal/mol for Oligo1 and -23.08 Kcal/mol for oligo 2. This is the best answer we can offer to the reviewer’s request, since the maximum binding free energy of DNAi individuals (much longer than the target strand) would include contributions from the unpaired bases. Indeed, the values give above are approached by the left tail of the distribution of Fig. 3a, which however includes DNAi self-energies.

The perfect complement binding free energy is now cited in the text as a reference for the dynamical range of DeltaG (p4 lines 151-152).

How is the number of captured DNA molecules quantified? Is 10^12 measured, estimated, or hypothesized?

The number of sequences was calculated from data obtained from 260 nm absorbance quantification. We have now added this information in the Methods, Selection Phase” section.

**Reviewer #2:**
Summary:In this manuscript, the authors introduced ADSE, a SELEX-based protocol to explore the mechanism of emergency of species. They used DNA hybridization (to the bait pool, "resources") as the driving force for selection and quantitatively investigated the factors that may contribute to the survival during generation evolution (progress of SELEX cycle), revealing that besides individual-resource binding, the inter- and intra-individual interactions were also important features along with mutualism and parasitism.Strengths:The design of using pure biochemical affinity assay to study eco-evolution is interesting, providing an important viewpoint to partly explain the molecular mechanism of evolution.Weaknesses:Though the evidence of the study is somewhat convincing, some aspects still need to be improved, mostly technical issues.Major:1. There are a few technical issues that the authors should clarify in the manuscript to make the analysis more transparent:(1.1) To my understanding, it is difficult to guarantee the even distribution of different species (individuals) in the initial individual pool. Even though the authors have shown in Fig. 2a that the top 10 sequences take up ~ 0% in the pool, it remains unclear how abundant these top and bottom representative sequences are, given the huge number of the pool (10E15). Can the author show the absolute number of these sequences in different quantiles? Please show both Oligo sets.:First, we thank the reviewer for both positive and critical comments that have guided us in reformulating or clarifying some messages of our work.

As for this specific point: 10E15 is a small number compared to 4^50 = 10E30, the number of possible sequences of length 30. Thus, we don’t expect more than one individual per sequence in the initial pool. However, sequencing requires a preparation amplification, which may lead to detecting a few sequences with more than one individual.

Specifically, in the initial pool of Oligo 1, the most abundant individual (of sequence GAACTAAAGGGGCGGTGTCCACTTGCCTGTAGTGGTTATCAGTCCGGTTG)has 3 copies.The 0.7% of the sequences has 2 copies, while the vast majority of strings (99.3% on a sample of about 1.5 x 10E6 sequenced DNAi) is present in one copy only. A similar situation holds for Oligo 2, with 4 DNAi present in 3 copies and the 0.8% of the sequences (in a pool of 2 x 10E6 DNAi) in 2 copies.

It is worth noticing that none of the 10 most abundant species in the last cycle is present in the sample. Indeed, the fraction of the pool which is sequenced is removed from the population that undergoes evolution (as now specified in p2, line 104). We specified in the text (p2, lines 69-70, p3 lines 94-96) the fact that in the initial pool no sequence is expected to be present in more than one individual.

(1.2) The author claimed that they used two different oligo sets (Oligo1 and Oligo2) in this study. It is unclear which data was used in the presentation. How reproducible are they? Similar to this concern, how reproducible if the same oligo set was used to repeat the experiment?

The oligo used in the main text was declared in Methods, Replica section. It is now declared also in the main text (p3 lines 106-108 and in the captions of Figure 2, Figure 3 and Figure 4). Reproducibility is addressed in: Figure 2-figure supplement 5; Figure 2-figure supplement 6; Appendix 2: Results of the experimental replica.

It should also be noted that two starting pools of random 50mers are necessarily disjoint sets for the same reason discussed in the previous answer: the probability of common sequences in two 10E15 selections from a 10E30 is negligibly small. Thus, it is expected that each time a new evolution experiment is started, different dominant sequences are found. However, the statistical properties of the DNAi pool during the evolution process of Oligo1 and Oligo2 are similar as discussed in Appendix 2 of the paper.

(1.3) PCR and illumina sequencing itself introduced selection bias. How would the analysis eliminate them? The authors only discussed the errors created during PCR cycles (page 3, lines 115-122). However the PCR itself would prefer to amplify some sequences over the others (e.g. with high GC content). Similarly, the illumina sequencing would be difficult to sequence the low complexity sequences. How would this be circumvented?

Yes, both PCR and Illumina sequencing have some known biases in the amplification process (e.g. sequencing of homopolymers or amplification of GC-rich sequences) that are intrinsic to the used techniques. Regarding PCR, we implemented a thermal protocol optimized for our chosen experimental setup, characterized by very short denaturation, annealing and amplification steps performed at high temperatures. Regarding Illumina sequencing, we can’t rule out a bias against specific sequences (e.g, homopolymers), which however should not be captured during the selection step, due to the design of the resource. Also, the libraries subjected to sequencing are characterized by a low complexity: according to the experimental design, the first and last 25 nucleotides are the same for all DNAi, the only differences being in the central 50 nt-long sequence. It is known that a low complexity library might encounter problems during sequencing due to the design of Illumina instruments: nucleotide diversity, especially in the first sequencing cycles, is critical for cluster filtering, optimal run performance and high-quality data generation. To overcome this limitation, the obtained libraries were run together with more complex and diverse library preparations: the ADSE sequences were about 1-2% of the total reads per run, corresponding to only a few million reads.

This discussion is now in Appendix 1, Intrinsic limitations of the molecular approach.

(1.4) Some DNA sequences would bind to the beads instead of the resource sequence coated on them. Should the author run the experiment using bead alone as a control?:We performed a negative control experiment in which we performed the “selection step” with bear beads, i.e. beads without with no DNA grafted on them. We then compared the results with the corresponding results of the original experiments on Oligo 1 and 2.

After 6 cycles, the most abundant sequence in the negative dataset has a relative occurrence of 0.05%, whereas the dominant strand in Oligo 1 and Oligo 2 has an abundance of 8% and 16%, respectively, i.e. 40-80 times larger.

This indicates that the drift due to non-specific binding (+ PCR amplification) is at least two orders of magnitude smaller than the selection induced by the affinity with the resource.

This part is now discussed in Appendix 1, Experimental controls.

1. It would be interesting to study the impact of environmental factors, for example, changing pH, salt concentration, and detergent. Would these factors accelerate/decelerate the evolution?

We agree that the approach we propose and the set of results we obtained call for further investigations. However, performing these additional experiments, which would require a minimum of 6 generations each, is a long and expensive job, which we have started and will not be completed in the near future. For this reason, given the already significant body of results we are presenting here, we prefer to keep this paper confined to the study of the evolution of a random DNAi population in the selected conditions and leave the exploration of new conditions, potentially opening new evolutionary scenarios, to a future contribution. In fact, our aim was to show that through our platform we can indeed observe fundamental elements of evolution in a non-biological system, which, in the set of chosen parameters, we do.

1. The concentration of individual oligo is apparently one of the most important factors in determining the interactions. In later cycles, some oligos become dominant, namely with extremely higher concentrations compared to their concentration in earlier cycles. This would definitely affect its interaction with resources, or self-interaction, or interaction with other oligos in the pool. However, the authors failed to discuss this factor, which may explain the exponential enrichment in later cycles.

We agree with the reviewer that this is an important point, but we disagree that we have not discussed it. We introduce the topic at the end of the “Null Model and Eco-evolutionary Algorithm”, where we comment on the change of the gamma parameter by saying that there must be a shift in the evolution process, first dominated by the interactions with the resources, and in later stages by some other factors (lines 227230) that we then discuss in “Self and mutual DNAi interactions are evolutionary drivers”. In this latter chapter and in the following, we indeed discussed the effects of mutual and self interactions between DNAi.

Indeed, a key point in our paper is the change in the gamma parameter necessary to match the IBEE model to experiments, as it is now more openly stated (p5 lines 217218 where we also mention figure 2-supplement 8 which clearly shows the necessity of a variable gamma). The two regimes enlightened by the gamma value must reflect a change in the competition for the resources and interactions among species. In the first generations, where the diversity of species is large (there are few strings for each species) and binding to the resources generally very week (small ), the affinity with the resource is the main driving force (fast growth of ), while mutual interactions remain too random to favor any species in particular. In the later cycles instead, when becomes large enough to provide a significant stability to the resource-binding of the majority of species, the dominating species compete more intensively on the basis of their structure and capacity of self-defense, parasitism and mutualism, a condition in which evolution affects more modifications in sequences than in .

Certainly, our understanding of this shift is based on statistical behavior and it is inferential, based on the study of specific DNAi described in the last part of the manuscript. For a better molecular model, more experiments with selected DNAi competing, cooperating or being parasitic would be necessary, with the final aim of defining a predictive fitness function. Alas, this requires months of further investigation.:

1. The author observed the different behaviors of medium *ω*� in early and late cycles, referring to Fig 2h. Using the IBEE model, they found out it is the change of gamma. However, the authors did not further discuss the molecular mechanism. It could be very interesting to understand the evolutionary change of these individuals.

This comment might be related to the previous one. It is true that our discussion and understanding of the whole process is statistical, and misses a molecular model to predict the value of gamma.

However, the specific behavior that the reviewer asks about (those in Fig. 2h) is not related to the change in gamma. Even if gamma remains as in the first part of the evolution (gamma = 3), the species with overlap between 6 and 10 would first grow in number and later decrease. Indeed, during the first cycles they have an advantage with respect to the majority of species with lower maximum overlap, a condition that favors their amplification. However, in the second stage of the evolution dominant species with a larger affinity emerge and outcompete the individuals of this class. We added a sentence in the text to clarify this point (p7 lines 227-229).

1. In Figure 2f, some high w become quite missing. Should the authors give some interpretation? It is not observed in cycle 12 though (panel e).

Such an effect is just due to under-sampling. In a pool of 10^n oligomers, any sequence with a given *ω*� with P(omega) < 10E-n will have a vanishing probability to appear in that sample.

At cycle 12 the overall number of sequenced strands is larger than at cycle 24, due to the growing presence of PCR by-products. Thus, the right tail of the cyan distribution at the last cycle is sampled with less accuracy than at cycle 12. We have added a sentence in the revised manuscript (p5 lines 177-178) to clarify this point.

1. It would be interesting to further explore if another type of selection resource is used, for example protein that binds to particular sequences, i.e. transcription factors. Previous studies have used a large amount of sequence-specific transcription factors to run SELELX. Since the data have existed there, why not explore?

This is an interesting suggestion: can we use data from “ordinary” SELEX favoring specific sequences to explore sequence evolution? Two limitations make us a bit skeptical on this path: first, the consensus sequences of DNA-binding proteins are rather short and typically target dsDNA rather than ssDNA; second, the free energy of interaction is known only for the consensus sequence but not for sequences with all possible mutations with respect to the consensus sequence, making very hard to develop any molecular understanding of the process.

Minor:1. There is no figure legend or in-text citation of Figure 2b.1. Please correct "⁃C" with "{degree sign}C" in lines 470, 471, 472, 477 et al.1. Typos and grammar issues should be corrected. Examples are shown below (but not limited to these only):mixed use of past and present tense.Line 152, "basis" should be "bases".Line 277, "a impediment" should be "an impediment"Line 278, "a major deadly threats" should be "major deadly threats":We are sorry for the mistakes, and we have corrected them. Many thanks to the reviewer!